# Direct translation of climbing fiber burst-mediated sensory coding into post-synaptic Purkinje cell dendritic calcium

Seung-Eon Roh[1,2,3,4,5†], Seung Ha Kim[1,2†], Changhyeon Ryu[1,2,3‡], Chang-Eop Kim[1,6‡], Yong Gyu Kim[1,2,3‡], Paul F Worley[5], Sun Kwang Kim[4]*, Sang Jeong Kim[1,2,3]*

[1]Department of Physiology, Seoul National University College of Medicine, Seoul, Republic of Korea; [2]Department of Biomedical Sciences, Seoul National University College of Medicine, Seoul, Republic of Korea; [3]Neuroscience Research Institute, Seoul National University College of Medicine, Seoul, Republic of Korea; [4]Department of Physiology, College of Korean Medicine, Kyung Hee University, Seoul, Republic of Korea; [5]Department of Neuroscience, School of Medicine, Johns Hopkins University, Baltimore, United States; [6]Department of Physiology, College of Korean Medicine, Gacheon University, Seongnam, Republic of Korea

**Abstract** Climbing fibers (CFs) generate complex spikes (CS) and $Ca^{2+}$ transients in cerebellar Purkinje cells (PCs), serving as instructive signals. The so-called 'all-or-none' character of CSs has been questioned since the CF burst was described. Although recent studies have indicated a sensory-driven enhancement of PC $Ca^{2+}$ signals, how CF responds to sensory events and contributes to PC dendritic $Ca^{2+}$ and CS remains unexplored. Here, single or simultaneous $Ca^{2+}$ imaging of CFs and PCs in awake mice revealed the presynaptic CF $Ca^{2+}$ amplitude encoded the sensory input's strength and directly influenced post-synaptic PC dendritic $Ca^{2+}$ amplitude. The sensory-driven variability in CF $Ca^{2+}$ amplitude depended on the number of spikes in the CF burst. Finally, the spike number of the CF burst determined the PC $Ca^{2+}$ influx and CS properties. These results reveal the direct translation of sensory information-coding CF inputs into PC $Ca^{2+}$, suggesting the sophisticated role of CFs as error signals.

*For correspondence:
skkim77@khu.ac.kr (SKK);
sangjkim@snu.ac.kr (SJK)

†These authors contributed equally to this work
‡These authors also contributed equally to this work

Competing interests: The authors declare that no competing interests exist.

## Introduction

Each Purkinje cell (PC), the sole cerebellar output neuron, receives strong excitatory inputs from the inferior olive (IO) through a single climbing fiber (CF), which innervates several PCs (*Eccles et al., 1966*). During cerebellar learning, the CF fires in response to unexpected sensory events to provide instructive signals to the PC, turning on $Ca^{2+}$ mediated plasticity mechanisms (*Hansel and Linden, 2000*; *Rancz and Häusser, 2006*). According to the Marr–Albus–Ito theory of learning, a CF-induced PC complex spike (CS) response is 'all-or-none' because IO stimulation generates seemingly binary responses in the PC (*De Schutter and Maex, 1996*; *Marr, 1969*). This notion has been prevailed as slice studies have show that a single CF stimulation induces similar EPSC (excitatory post-synaptic current) above a certain stimulus intensity (*Konnerth et al., 1990*) and is enough to induce parallel fiber (PF)-PC synapse long-term depression (LTD) (*Ito and Kano, 1982*), in which the level of learning is determined by the range of PF excitation (*Reynolds and Hartell, 2000*). This means that the CF's impact has little variation in strength and does not carry quantitative information, although CF input, by controlling the $Ca^{2+}$ transient in PC, could critically affect PF-PC LTD (*Ohtsuki et al., 2009*). Instead, it is generally accepted that the magnitude of learning is affected by how many invariant CFs synchronously fire (*Squire, 2009*).

However, the so-called 'all-or-none' property of CF error signal becomes questionable when it comes to in vivo conditions, since the CF's bursting properties have been described (*Mathy et al., 2009*) and a single non-burst stimulation of CF failed to induce associative learning (*Rasmussen et al., 2013*). In line with this, the previously unknown complexity of the error signal has received attention (*Najafi and Medina, 2013*). The CF burst reflected the olivary oscillation frequency and the number of spikes ranged from 1 to 3 in anesthetized animals (*Mathy et al., 2009*). Also, burst activity seemed to be affected by certain types of visual stimulation (*Maruta et al., 2007*). Furthermore, longer CS duration, presumably resulting from the longer burst, was associated with enhanced cerebellar motor learning (*Yang and Lisberger, 2014*).

It is thus highly interesting to determine whether and how a CF itself encodes quantitative information of unexpected sensory events, such as differential stimulus intensities, and transmits the error signals in a graded manner to the PC in awake animals. One recent report described the sensory-driven enhancement of PC $Ca^{2+}$ signals, in which the authors only observed post-synaptic PC $Ca^{2+}$ transients, which they claimed non-CF components affect (*Najafi et al., 2014b*). Although CF $Ca^{2+}$ activity was described in vitro (*Nishiyama et al., 2007*), so far, sensory coding by the CF can hardly be investigated because sensory inputs disappear in ex vivo preparation and the direct recording of the CF axon has been technically infeasible to obtain in vivo except for recent studies (*Gaffield et al., 2018*; *Gaffield et al., 2019*). Using genetically encoded $Ca^{2+}$ indicators (GECIs) and in vivo 2-photon microscopy imaging of CF and/or PC populations in awake mice, here we show CF burst-mediated sensory coding and its direct representation in a post-synaptic PC. At rest, the CF $Ca^{2+}$ activity exhibited highly variable strength and synchrony. Electrophysiological analysis indicated a direct correlation between CF $Ca^{2+}$ amplitude and the number of spikes in the CF burst. Employing unexpected sensory stimuli, we revealed that CF $Ca^{2+}$ signal encodes sensory stimulus intensity, just as PC $Ca^{2+}$ activity does. Our dual-color simultaneous imaging of CF and PC $Ca^{2+}$ signals also indicated a linear correlation between CF and PC responses in awake animals, suggesting that pre-synaptic CF inputs carrying sensory information strongly contribute to post-synaptic PC responses.

## Results

### CF $Ca^{2+}$ activity is highly variable in strength and associated with the number of spikes in the burst

CF activities have been indirectly inferred from the signatures of CS in PCs by attached or field potential recordings, which are contaminated with local circuit activities such as those of the PF or molecular interneuron (MLI). In this study, 2-photon $Ca^{2+}$ imaging with GECI expression enabled us to directly visualize in vivo CF activity. We injected an adeno-associated viral (AAV) vector into the IO, which expresses GCaMP6f using *Camk2a* promoter (*Figure 1a*). Histological characterization reveals strong green fluorescence in bilateral IOs (*Figure 1a*). The neuronal somas in the brain stem and the axon fibers in cerebellar white matter appeared green (*Figure 1a2-3*). CF axon buttons and shafts were also observed in the molecular layer (*Figure 1a4*).

To observe CF $Ca^{2+}$ activity, we created a cranial window on the cerebellar cortex of the lobule IV/V vermis while injecting AAV-*Camk2a*-GCaMP6f in IO. Three weeks later, we performed 2-photon $Ca^{2+}$ imaging in awake, head-fixed mice on a disk treadmill (*Figure 1b*). The coronal projection image of the z-stack indicates the strong fluorescence of CFs that terminate in the cerebellar cortex (*Figure 1b1*). The z-projection image of molecular layer clearly shows CF axonal varicosities in the CFs (*Figure 1b2* from the box in 1b1). For accurate identification of simultaneously firing pixels of CF $Ca^{2+}$ and reliable signal/event detection, we have utilized Suite2p, an open-source $Ca^{2+}$ imaging tool, which allows detection of the $Ca^{2+}$ signals from axonal varicosities (*Marius Pachitariu et al., 2016*). Suite2p efficiently identified CF varicosities where ones with correlation coefficient of 0.85 were merged in its graphical user interface, while ROIs with low signal-to-noise level are discarded (*Figure 1c*). Similar to a previous report about CF-evoked PC activity (*Ozden et al., 2012*), the resting state CF firing frequency was 0.85 Hz ±0.24 SEM and the CFs population also showed highly synchronous activity within an imaging field of 208 µm-wide (mediolaterally) (*Figure 1e*). As the $Ca^{2+}$ synchrony of nearby PC dendrites is known to decline through mediolateral separation (*Ozden et al., 2009*; *Schultz et al., 2009*), the synchrony level gradually fell as a function of mediolateral distance (*Figure 1f* and *Figure 1—source data 1*).

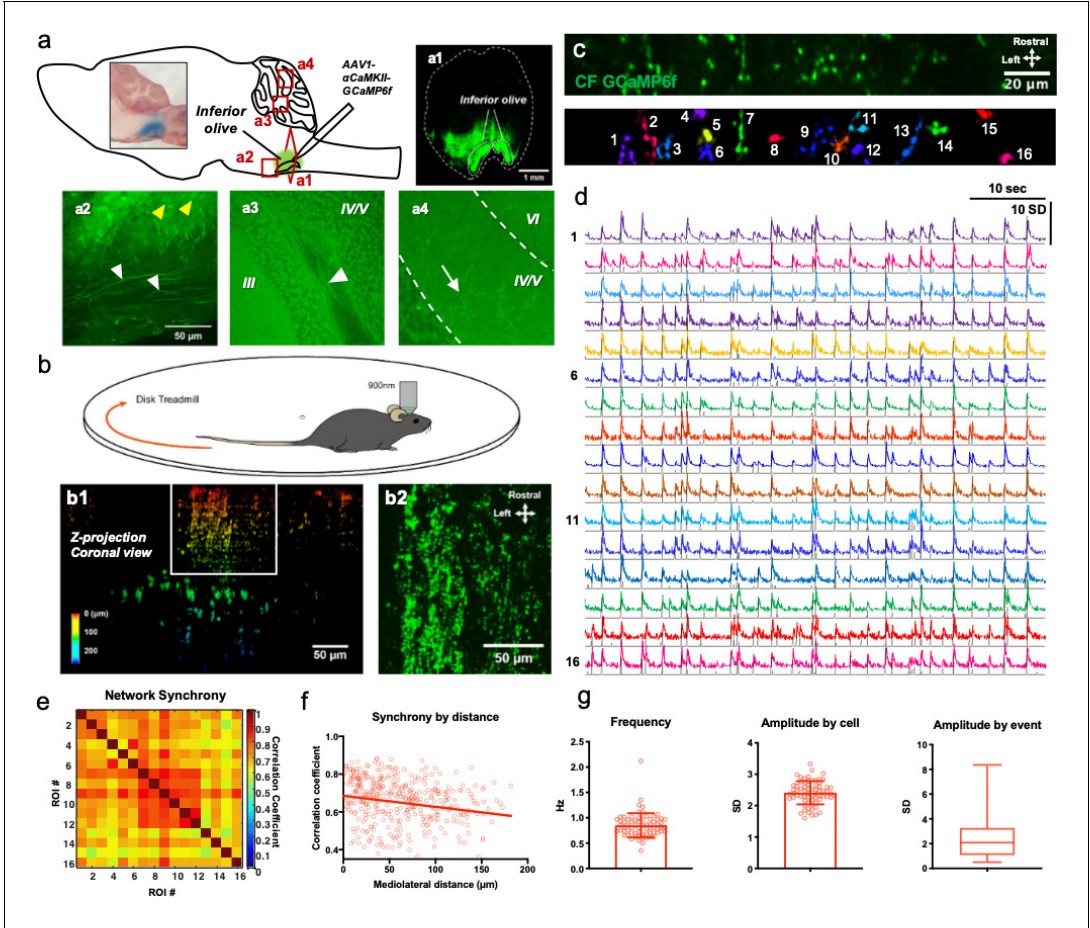

**Figure 1.** The CF Ca$^{2+}$ activity is highly variable in awake mice in vivo. (**a**) Schematic diagram of the IO viral injection. A coronal section view of the brain stem region of a brain in which GCaMP6f was expressed for 3 weeks (**a1**). GCaMP6f expressed soma and projecting axons (yellow and white arrowhead, respectively) in a brain stem region including the IO (**a2**). GCaMP6f-expressed axons (white arrowhead) within white matter (**a3**). GCaMP expressed CF varicosities (white arrow) within the molecular layer of lobules IV/V (**a4**). (**b**) A schematic diagram of two-photon microscopy of awake mice on a disk treadmill. A coronal view of the z-stack projection image of CFs expressing GCaMP6f in the cerebellar cortex (**b1**). A dorsal view of the z-stack image (maximum projection image) of white box regions of b1, which represent the molecular layer (**b2**). (**c**) An example of ROI detection of CF varicosities using the Suite2p. The field of view is 512 × 64 pixels. (**d**) Resting-state GCaMP6f intensity traces the 16 ROIs over 60 s with event detection plot (grey lines). Intensities were expressed as standard deviation as signals were z-score normalized. (**e**) An example matrix of correlation coefficients among every pair of the 16 ROIs, with the Pearson correlations described by colored intensity. Right: A scale bar for correlation coefficient. (**f**) The correlation coefficient among the CFs in terms of the mediolateral distance between each pair of all ROIs. n = 397 pairs of CFs, R$^2$ = 0.034. (**g**) Average frequency (0.85 Hz ± 0.24 SEM, n = 69 varicosities), amplitudes by cell (2.41 SD ± 0.045 SEM, n = 69 cells), and amplitudes by event (2.349 SD ± 1.49 SD, n = 3481 events, N = 5 mice).

The online version of this article includes the following source data for figure 1:

**Source data 1.** Statistics for spontaneous Ca2+ activity.

Strikingly, CF amplitudes were substantially variable (2.349 ± 1.49 SD, *Figure 1g* and *Figure 1— source data 1*) with maximum being up to 8.366, while the average Ca$^{2+}$ amplitudes per cell converged to 2.41 SD ±0.045 SEM. *Mathy et al., 2009* recorded in vitro CF axonal activity and described its bursting property, which encodes olivary oscillation. Also, EPSC numbers in PC, presumably evoked by the spike number of the CF burst varied between 1 and 3 under anesthesia. Because the CF Ca$^{2+}$ activity we observed here was also variable, we asked whether CF Ca$^{2+}$ activity is related to the number of spikes in the burst. In cerebellar slices prepared from the mice expressing GCaMP6f in IO, CFs were directly stimulated at the granule cell layer with 1 to 9 spikes in 400 Hz bursts (*Figure 2a,b*). Our data indicate that increasing the number of spikes in the burst stimulations accordingly augments the CF Ca$^{2+}$ amplitude, which is saturated with a burst of seven spikes (*Figure 2c–d* and *Figure 2—source data 1*), showing that the variability of CF Ca$^{2+}$ activity is

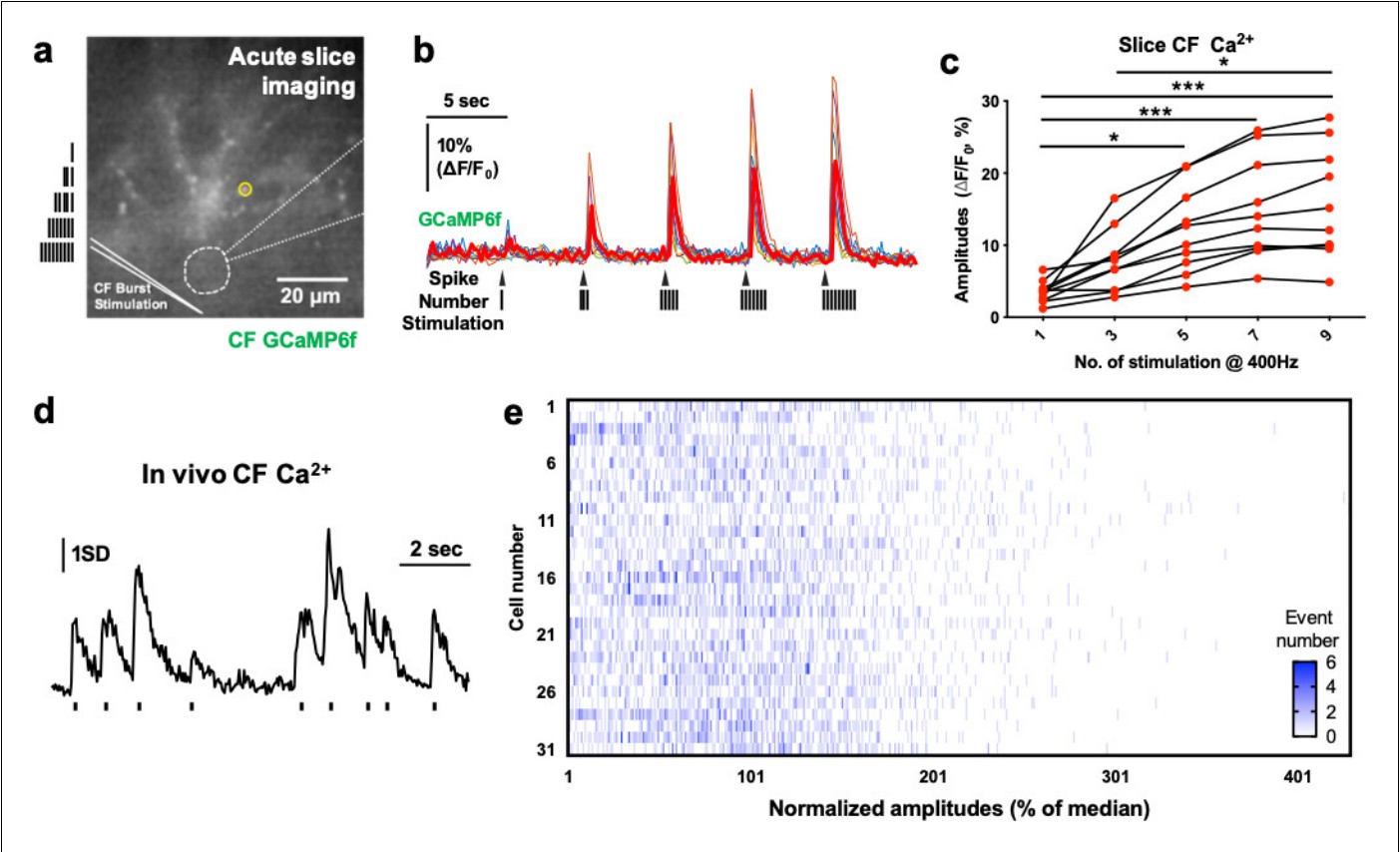

**Figure 2.** The variable CF Ca$^{2+}$ activity encodes the spike number in the burst. (a) The sagittal slice image of GCaMP6f-expressed CF in the cerebellar PC and molecular layer. Five responding axonal varicosities (<2 µm) were selected (as indicated by yellow circles) and averaged for each cell's traces. The approximate PC soma was marked with a thick white dashed line. The patch pipette was depicted as a thin white dashed line. (b) Burst stimulation–evoked CF GCaMP6f intensity plots of the 30 s recordings of nine independent CFs from three mice. The thick red trace represents the averaged trace. (c) Quantification of the burst-evoked GCaMP6f amplitudes. One-way ANOVA followed by Bonferroni test: ***p<0.001, *p<0.05. (d) Example trace of in vivo CF Ca$^{2+}$ imaging showing a sampling of events for amplitude distribution analysis in e. Only the first-peak amplitudes out of 0.5 s window were analyzed. The short lines below the trace represent the sampled events. (e) Amplitude distribution of 31 CFs from four independent 3 min recordings of 2-photon Ca$^{2+}$ imaging in three animals. The first-peak amplitudes were normalized with the median values to display all data and were presented as a heat map histogram, in which x and y represent normalized amplitudes and cell numbers, respectively. The color map scale shows the number of events.

The online version of this article includes the following source data for figure 2:

**Source data 1.** The CF Ca2+ activity is highly variable in awake micein vivo.

positively correlated with the number of spikes per CF burst. To support this notion, we performed amplitude histogram analysis with in vivo CF Ca$^{2+}$ amplitudes of 3 min recordings. Only first-peak amplitudes within 0.5 s windows were gathered since closely following events can exert an additive effect on amplitudes (*Figure 2d*). Interestingly, the distribution appeared discrete, with several amplitude clusters existing (*Figure 2e* and *Figure 2—source data 1*), possibly driven by the number of spikes in the burst. Hence, the huge variability in CF Ca$^{2+}$ amplitude may be caused by the variable number of CF bursts.

## The spike number of CF bursts encodes graded sensory information

Considering the variability of CF Ca$^{2+}$ by spike number (*Figure 1g*), we asked whether the spike number in the CF burst could encode the intensity of natural sensory stimuli by employing periocular air-puff stimulation (*Figure 3a*), which triggers trigeminal CFs projecting to the paravermal lobule V regions (*Najafi et al., 2014a*). We tested the two strengths of stimulation sets that were reported to differentially modulate PC dendritic Ca$^{2+}$ responses (*Najafi et al., 2014a*). In this study, we only

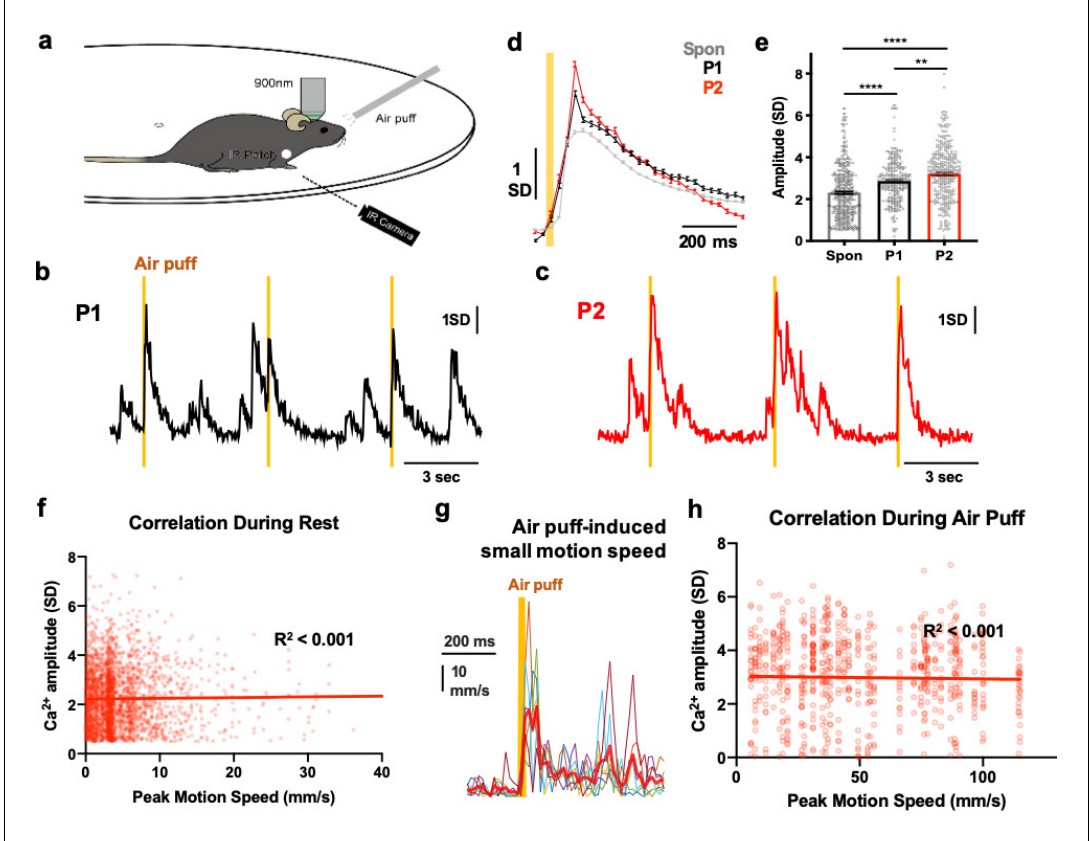

**Figure 3.** Sensory coding by CF Ca$^{2+}$ signals. (a) A schematic showing the 2-photon imaging with periocular air-puff stimulation and animal motion monitoring by IR camera. (b-c) Representative CF Ca$^{2+}$ traces for spontaneous and 30 ms periocular air puffs (orange column) of pressure 1 (P1, b) and 2 (P2, c). (d-e) Averaged CF Ca$^{2+}$ traces (d) and amplitudes (e) of spontaneous and air-puff responses of P1 and P2. n = 448, 283 and 346 CF Ca$^{2+}$ events in 4 (P1) and 6 (P2) independent 1 min imaging sessions from four mice, respectively. One-way ANOVA followed by Tukey's test: **p<0.01, ***p<0.001, **** p<0.0001. (f) Correlation analysis between CF Ca$^{2+}$ amplitudes versus the peak speed of forepaw movement at rest. n = 2926 pairs from four recordings in two mice, R$^2$ <0.001. (g) Representative traces of the speed of forepaw twitch-like movement during periocular air-puff stimulation. Air puff–to–motion onset time = 23.2 ± 2.9 SEM ms, air puff–to–peak motion speed time = 57.6 ± 5.8 SEM ms. (h) Correlation analysis between CF Ca$^{2+}$ amplitudes versus the peak motion speed during air-puff stimulation. n = 760 pairs from four recordings in two mice. R$^2$ <0.001. The online version of this article includes the following source data and figure supplement(s) for figure 3:

**Source data 1.** Sensory coding by CF Ca$^{2+}$ signals.

**Figure supplement 1.** Minimal motion artifacts for imaging during air-puff stimulation.

**Figure supplement 2.** Sensory coding by Purkinje cell dendritic Ca$^{2+}$ activity.

**Figure supplement 2—source data 1.** Sensory coding by Purkinje cell dendritic Ca$^{2+}$ activity.

analyzed data at resting state and excluded images in which the animals are walking or running since locomotion produces complex Ca$^{2+}$ transients. As for the strength, CF Ca$^{2+}$ amplitude was significantly enhanced with lower pressure (P1) than with spontaneous responses and further increased in higher pressure (P2) (*Figure 3b–e* and *Figure 3—source data 1*). Here, neither P1 nor P2 air-puff stimulation caused motion artifacts in our experimental condition (*Figure 3—figure supplement 1*). These data indicate that the CF Ca$^{2+}$ amplitude—the spike number of the burst, in other words—conveys information about sensory stimuli with differential strength.

CF inputs to the cerebellum have also been known to be associated with body movements (*Ozden et al., 2012*) and could be evoked even by small movements in the paravermal area of lobule V (*Rushmer et al., 1976*). Since CF Ca$^{2+}$ has movement-related activities in our experiments, we tested whether the spike numbers encoded in CF Ca$^{2+}$ amplitudes are related to movements by correlation analysis between CF Ca$^{2+}$ amplitudes and the strength of the small movement. The animal's motion speed was acquired using high-speed IR camera by tracking the IR-reflective patch on the

forepaw at rest and during air-puff stimulation (*Figure 3a*). At rest, CF Ca$^{2+}$ events were not correlated with the peak motion speed (R$^2$ <0.001, *Figure 3f* and *Figure 3—source data 1*). Air-puff stimulation generally induced small twitch-like movement (*Figure 3g* and *Figure 3—source data 1*), while it was not correlated with CF Ca$^{2+}$ amplitudes either (R$^2$ <0.001, *Figure 3h*). Although we did not examine any other movements related to air-puff stimulation, such as orofacial movement, we suggest that the CF Ca$^{2+}$ activity could differentiate the graded sensory stimuli but not movement strength under the assumption that forepaw motion represented startle movement during the air-puff stimulation.

Next, we sought to determine whether our set of ipsilateral periocular air-puff stimuli induced graded responses in PC dendritic Ca$^{2+}$, as previously reported (*Najafi et al., 2014b*). For the specific expression of GECI in PCs, we used *Pcp2*-cre transgenic (TG) mice, of which cerebellar vermis was targeted for cre-dependent jRGECO1a expression (*Figure 3—figure supplement 2a–b* and *Figure 3—figure supplement 2—source data 1*). The PC dendritic Ca$^{2+}$ activity was similarly detected with Suite2p. Sensory stimuli were shown to enhance PC Ca$^{2+}$ responses, as compared to spontaneous responses, and were graded with different pressure strengths (*Figure 3—figure supplement 2c–d*). The results collectively suggest that the spike number of CF bursts reflects sensory strength and may direct the PC-mediated strength-dependent sensory coding.

## Direct translation of spike number in the CF burst by PC dendritic Ca$^{2+}$ signals

Although the aforementioned data suggest that both the CF and PC process sensory information, these results do not ensure their direct correlation. Hence, we set out to simultaneously record the pre- and post-synaptic Ca$^{2+}$ activity. In *Pcp2*-cre TG mice, AAV-*Camk2a*-GCaMP6f was injected into the IO (*Kimpo et al., 2014*), and the cre-dependent expression of jRGECO1a—the sensitive red color Ca$^{2+}$ indicator (*Dana et al., 2016*)—was achieved (*Figure 4a*). Dual-color imaging was performed using GFP and RFP filters under 1000 nm two-photon laser excitation, which revealed the structure of CF axon varicosities and PC dendrites located adjacent to them (*Figure 4b*). The CF-PC pairs were readily identifiable by their proximity (*Figure 4b*). To check whether signal bleed-through existed between the two channels, dual-channel imaging was performed in the cerebellums expressing only CF-GCaMP6f and PC-jRGECO1a, respectively. Even though some very strong signals of CF-GCaMP6f varicosities were visible in the red channel, no obvious spectral overlaps existed in selected ROIs (*Figure 4—figure supplement 1a*). Also, the transmission of jRGECO1a signals into EGFP filters was negligible (*Figure 4—figure supplement 1b*). Furthermore, Ca$^{2+}$ signal processing with Suite2p should remove such negligible bleed-throughs since Suite2p detects fluorescence traces with a high signal-to-noise ratio (*Marius Pachitariu et al., 2016*).

As shown in *Figure 4c*, signal traces and the detected events from the two CF-PC pairs revealed a highly synchronous activity (*Video 1*). To determine whether CF Ca$^{2+}$ activity covaries with PC Ca$^{2+}$ activity, we performed correlation analysis in terms of signal and paired amplitudes. For signal correlation analysis, the correlation coefficient between the CF and PC Ca$^{2+}$ signals was computed. The CF and the identified PC dendrites spatially located in the proximity (<1 μm) with the CF were regarded as 'paired' CF-PC (*Figure 4b,d*), for which the correlation coefficient reveals a close relationship (R = 0.605 ± 0.027 SEM). But the correlation was significantly lower in 'unpaired' CF-PC, where PC dendrites were remotely located from the CF ROIs (R = 0.443 ± 0.032 SE, *Figure 4d* and *Figure 4—source data 1*). The 'unpaired' CF-PC pairs were further categorized as 'neighboring unpaired' and 'distant unpaired' if at least 30 μm apart from each other. The distantly located CF-PC showed significantly lower signal correlation (0.327 ± 0.057 SE) compared to neighboring ones (0.498 ± 0.032 SE, *Figure 4d,e* and *Figure 4—source data 1*). Next, we analyzed the correlation of CF-PC amplitude pairs that were collected where the CF Ca$^{2+}$ events

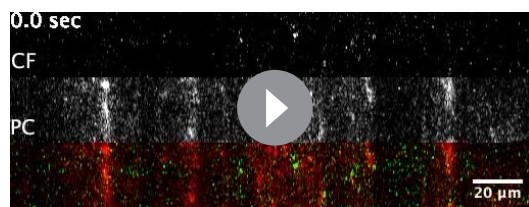

**Video 1.** CF-PC dual-calcium imaging. 60 s 32 Hz time-lapse movie for CF-PC dual imaging by two-photon microscopy using 1000 nm excitation. CF, PC, and merged images from the top to the bottom. Time scale is indicated in upper left and the scale bar is indicated lower right as 20 μm.
https://elifesciences.org/articles/61593#video1

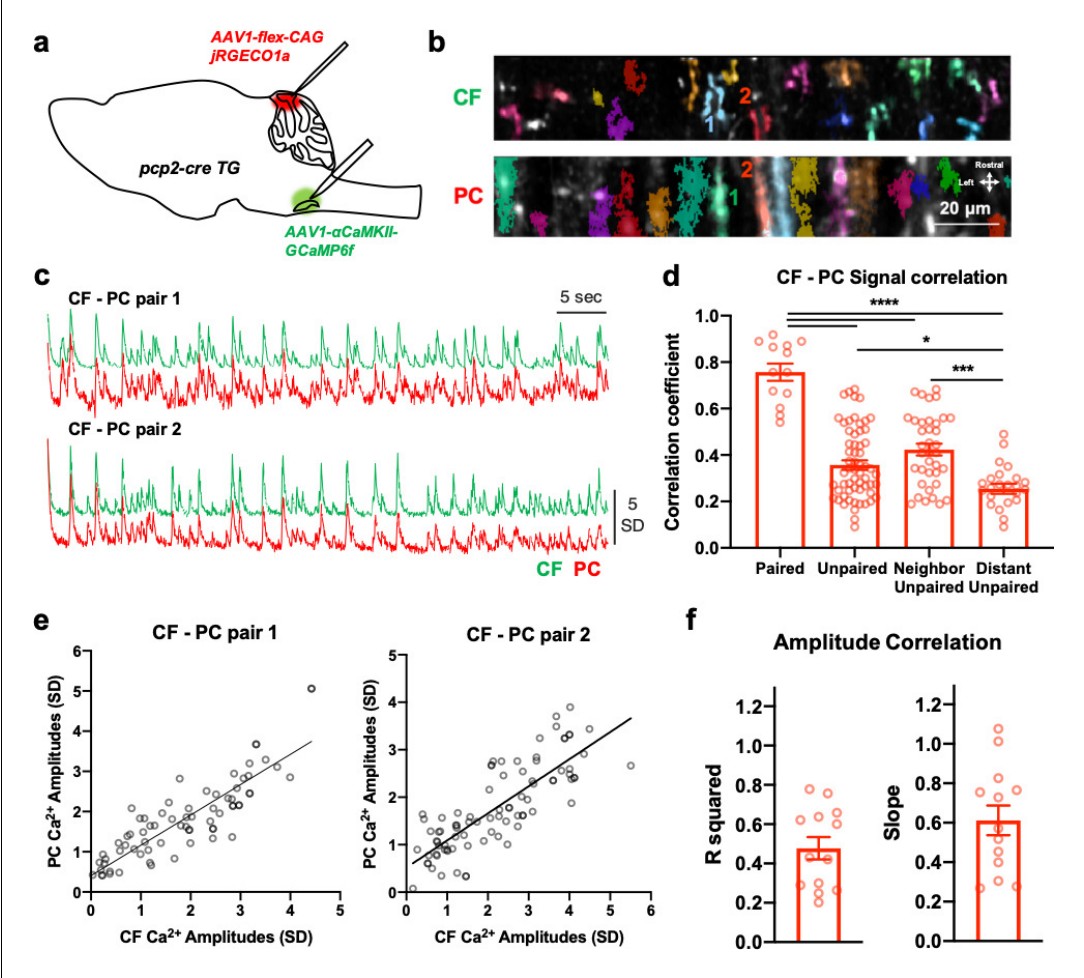

**Figure 4.** The direct translation of spike number for each CF burst by post-synaptic PC $Ca^{2+}$ response. (**a**) A schematic that shows the dual expression of $Ca^{2+}$ indicator jRGECO1a and GCaMP6f in PC and CF, respectively. (**b**) ROI detection by Suite2p and average projection images of CF and PC dual-calcium imaging in the cerebellar cortex (50 µm from dura). Two examples of CF-PC pairs are indicated with number. (**c**) Traces of the two CF-PC pairs (1 and 2) from b. (**d**) Correlation coefficients of the signals are shown in four conditions which include 'paired', 'unpaired', 'neighboring unpaired' and 'distant unpaired'. n = 13 (paired), 61 (unpaired), 37 (neighboring unpaired), and 24 (distant unpaired) CF-PC pairs. One-way ANOVA followed by post hoc Tukey multiple comparisons test. ****$p<0.0001$, ***$p<0.001$, *$p<0.05$. (**e**) A representative non-linear regression analysis of $Ca^{2+}$ amplitudes in two pairs of CF-PC shown in c. n = 73 (pair 1) and 85 (pair 2) events. $R^2$ = 0.756 (pair 1) and 0.658 (pair 2). (**f**) $R^2$ values (left) and slope (right) for CF-PC pair $Ca^{2+}$ amplitude-correlation analysis with 13 CF-PC pairs. Average $R^2$ = 0.48 ± 0.06 SEM. Average slope = 0.61 ± 0.08 SEM. Data are from five independent recording sessions of three mice.

The online version of this article includes the following source data and figure supplement(s) for figure 4:

**Source data 1.** The direct translation of spike number for each CF burst by post-synaptic PC Ca2+ response.

**Figure supplement 1.** Limited signal bleed-through during simultaneous dual-color recording.

occurred. Linear regression analysis with the two CF-PC pairs revealed significant correlation (*Figure 4e* and *Figure 4—source data 1*). Such a prominent correlation was consistent with 13 pairs (Average $R^2$ = 0.48 ± 0.06 SEM, *Figure 4f* and *Figure 4—source data 1*). Thus, the results suggest that CF $Ca^{2+}$ signals are directly translated into post-synaptic PC $Ca^{2+}$ signals in awake animals.

## The spike number of CF bursts directly affects the amplitudes of PC dendritic $Ca^{2+}$ response

*Mathy et al., 2009* reported that varying the number of spikes in CF burst affects the number of spikes in post-synaptic PCs. Hence, we tested whether the different spike numbers of CF bursts cause graded $Ca^{2+}$ spike amplitudes in post-synaptic PC with ex vivo $Ca^{2+}$ imaging (*Figure 5a*). The

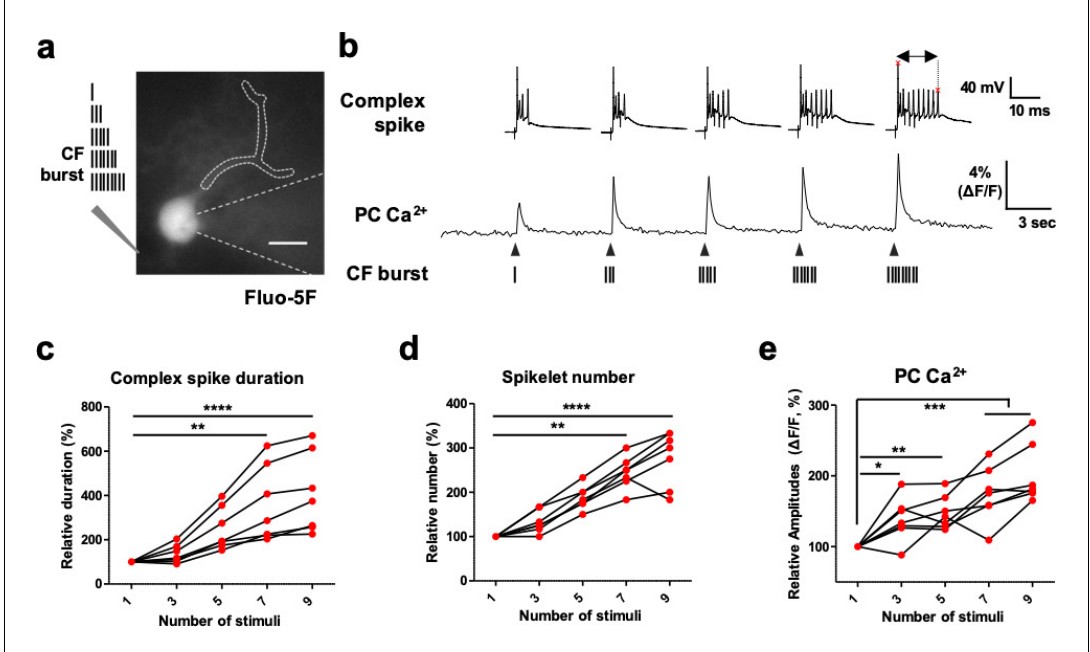

**Figure 5.** The CFs' spike number-dependent Ca$^{2+}$ influx in PC dendrites. (**a**), Representative image of a PC filled with the low-affinity Ca$^{2+}$ dye Fluo-5F taken with whole-cell recording. The schematics on the left describe the number of CF stimuli (1, 3, 5, 7, and 9) at 400 Hz. **b**, Representative aligned traces of CS recordings and PC Ca$^{2+}$ traces measured by Fluo-5F in response to the indicated numbers of 400 Hz CF stimuli (1, 3, 5, 7, and 9). The length between the two red asterisks represents the duration of CS. c–e, CS duration (**c**), spikelet numbers, (**d**) and amplitudes of the post-synaptic PC Ca$^{2+}$ transient (**e**) in response to different number of spikes in the CF burst stimuli. n = 7 recordings of seven independent cells from three mice. One-way ANOVA followed by Bonferroni test: *p<0.05, **p<0.01, ***p<0.001, and ****p<0.0001.

The online version of this article includes the following source data for figure 5:

**Source data 1.** The CFs' spike number-dependent Ca2+ influx in PC dendrites.

low-affinity Ca$^{2+}$ dye Fluo-5F was loaded after making the whole-cell at the PC soma. The paired CFs were stimulated at 400 Hz while whole-cell recording and Ca$^{2+}$ imaging were performed. Interestingly, a higher number of burst stimuli induced greater amplitudes in the post-synaptic PC dendritic Ca$^{2+}$ response (*Figure 5b,e* and *Figure 5—source data 1*). Also, the duration of CS and the spikelet number were significantly enhanced the number of CF stimuli increased (*Figure 5b–d* and *Figure 5—source data 1*). These results strongly indicate that the spike number of CF bursts directly affects the PC CS properties as well as Ca$^{2+}$ amplitudes, suggesting powerful presynaptic governance in PC Ca$^{2+}$-mediated sensory coding.

## Discussion

In our study, we made a series of novel observations demonstrating the significant role of CF input in cerebellar sensory coding. First, Ca$^{2+}$ imaging in CF axon varicosities in awake mice showed great variability in a resting-state activity. The ex vivo experiments revealed that Ca$^{2+}$ activity directly reflects the number of spikes in CF bursts. Also, by applying air puffs as sensory stimuli, we found that CF bursts convey quantitative sensory information, just as PC Ca$^{2+}$ signals do. Further, CF-PC dual-color Ca$^{2+}$ imaging revealed a systemic correlation between pre- and post-synaptic activity during rest. Finally, the number of spikes in the CF burst linearly affected the CS properties and Ca$^{2+}$ influx in PCs. These results suggest that PC dendritic Ca$^{2+}$ activity and its sensory coding process are largely governed by the sophisticated control of presynaptic CF inputs.

### The substantial variability of CF Ca$^{2+}$ activity stems from its burst activity

The classical view of olivo-cerebellar transmission is that PC CS is an 'all-or-none' response (*Eccles et al., 1966*). The CF activity itself has also been regarded as having a binary property, but

the poor signal-to-noise problem is compensated for by pooling the activity of multiple CFs (*Najafi and Medina, 2013*). This was further supported by observations that synchronous activity of PC CS is related to cerebellar information processing, such as movements and sensory stimuli (*Ozden et al., 2012*). Hence, it has been thought that more significant information is processed when more CFs are activated. However, it remains unresolved whether individual CFs encode parametric information of sensory input (*Squire, 2009*).

The description of CF burst activity in vivo has made the properties of individual CFs appear more complicated than previously thought (*Mathy et al., 2009*). In that study, the recorded number of spikes in the CF burst varied from 1 to 3 in anesthetized animals, making it possible that the spike number reflects the degree or types of certain information (*Mathy et al., 2009*). Yet, the authors concluded that the spike number may not carry the strength or intensity of sensory inputs, in support of a study in which cat IO neurons showed a unitary spike in response to the stimulation of afferent inputs to IO (*Crill, 1970*). However, such studies rely on results from anesthetized animals and the physiological role of axon burst has remained unanswered. In our study, we overcame the difficulty of direct CF recording by employing two-photon $Ca^{2+}$ imaging and GECI expression in CFs. Also, since general anesthesia disrupts normal neuronal firing (*Greenberg et al., 2008*), we performed imaging in awake animals. The open-sources software, Suite2p, enabled us to successfully detect individual CF varicosities and their $Ca^{2+}$ signals (*Figure 1c–d*). The paired correlation fell off with increased mediolateral distance (*Figure 1f*), which is similar with the PCs' correlation (*Schultz et al., 2009*). The high variability of CF $Ca^{2+}$ amplitudes in awake mice, even during the resting state, was very interesting (*Figure 1d,g*). Although small sounds or movement could induce CF firing (*Ozden et al., 2012*; *Rushmer et al., 1976*), the maximum amplitude was 8.366 SD, suggesting a highly variable range of fold-change (*Figure 1g*). This result is in line with a study that reported the number of excitatory post synaptic potentials (EPSPs) of PCs ranges from 1 to 5 during spontaneous activity (*Maruta et al., 2007*). Also, a recent study reported the variability of CF $Ca^{2+}$ activity in the Crus II lobule (*Gaffield et al., 2019*). Importantly, we revealed that the CF $Ca^{2+}$ amplitude was directly regulated by the number of spikes in the CF axon burst stimulation at 400 Hz (*Figure 2a–c*), which suggests that the $Ca^{2+}$ amplitudes are a readout of the degree of bursts. The physiological number of spikes likely ranges from 1 to 6 or seven in an awake state, since the ex vivo CF $Ca^{2+}$ responses are saturated in a burst of 7 spikes. The discrete amplitude distribution of in vivo signals further supports the dependency on the spike number in the burst (*Figure 2e*). Thus, the great variability of CF $Ca^{2+}$ signals in awake mice tells us that the spike number in the CF burst also varies even in the resting condition.

## CF bursts convey sensory information to post-synaptic PCs

Several reports have shown that PC dendritic $Ca^{2+}$ spikes convey sensory information (*Kitamura and Häusser, 2011*; *Najafi et al., 2014a*; *Najafi et al., 2014b*). We also confirm that PC $Ca^{2+}$ amplitudes differentiate the strength of periocular air-puff stimuli (*Figure 3—figure supplement 2*). If the sensory coding of a PC is derived from presynaptic CF activity, then the CF should differentiate the sensory stimuli and its strength. Here, we observed how sensory input enhanced CF $Ca^{2+}$ response which was further enhanced with stronger stimuli (*Figure 3*). This is strong evidence for the critical role of presynaptic CF input in shaping PC $Ca^{2+}$ response during unexpected sensory events. Also, the results from dual-color $Ca^{2+}$ recording of CF axons and PC dendrites in resting-state revealed that the signal and amplitudes of CF-PC pairs have robust correlations (*Figure 4*), suggesting that PC $Ca^{2+}$ activity is strongly driven by CF burst activity.

It seems that PF and CF inputs converge onto the same PC during sensory events (*Apps and Garwicz, 2005*). This allows some to argue that PF may contribute to the sensory coding of CF-evoked PC $Ca^{2+}$ amplitudes by showing that small supralinear non-CF inputs were detectable in the absence of CF-induced response (*Najafi et al., 2014b*). However, a recent study reported that silencing CF activity completely abrogated PC $Ca^{2+}$ activity (*Gaffield et al., 2019*), and it is also unlikely that PF-evoked $Ca^{2+}$ activity is dendrite-wide as CF-evoked responses are (*Kitamura and Häusser, 2011*). Further, the so-called 'sensory-evoked non-CF inputs' do not account for the tight coupling of CF-PC activity even during the resting condition (*Figure 4d*). We presume that such non-CF contributions in CF-evoked PC $Ca^{2+}$ response are scarce, and PC $Ca^{2+}$ activity—during resting or sensory processing—is mostly determined by CF burst activity. On the other hand, however, CF and PC $Ca^{2+}$ signals were not perfectly correlated. Hence, CF activity may serve as a trigger with qualitative

information for post-synaptic response, and other factors, such as PC intrinsic properties (*Kitamura and Häusser, 2011*), PF activity-mediated depolarization (*Wang et al., 2000*) and noradrenergic pre-synaptic control (*Carey and Regehr, 2009*), may participate to form an even more complex shape of PC $Ca^{2+}$ and CS responses.

## Physiological impacts of CF pre-synaptic governance over PC activity

What is the meaning of CFs having such powerful and fine control over PCs? First, the strength of the cerebellar output to the deep cerebellar nucleus (DCN) can be regulated at the level of the CFs' input magnitudes. It has been thought that an increased population of so-called binary CF activation will generate stronger PC-mediated outputs (*Squire, 2009*). However, our data suggest that individual CFs can provide a differential magnitude of inputs onto PCs, depending on the number of spikes in their burst (*Figure 5*). Considering the high synchrony of CF population activity (*Figure 1g*), they tend to fire simultaneously like PCs do (*Kitamura and Häusser, 2011*; *Ozden et al., 2012*; *Tsutsumi et al., 2015*). On top of that, the graded amplitudes of synchronously firing CFs can help generate a diverse range of cerebellar outputs. Second, the variable CF activities will present more sophisticated error signals to the PC during learning. *Yang and Lisberger, 2014* presumed that the CS duration is the critical determinant for the degree of learning, as expected by recording CSs in monkey undergoing smooth pursuit learning. They found that the CS duration tends to be longer in the first 30 trials out of 100. This is also in line with a study showing that the degree of IO stimulation determines the direction of learning (*Rasmussen et al., 2013*). We provide direct evidence that CS duration, spikelet number, and PC dendritic $Ca^{2+}$ are all critically affected by the number of CF bursts (*Figure 5*), suggesting that stronger or newer experiences will generate longer CSs with more spikelet number and $Ca^{2+}$ influx (*Figure 3*). Thus, the CF $Ca^{2+}$ activity may be strong at first and weaken over time during cerebellum-dependent learning such as during eye-blink conditioning. Also, limiting the spike number to one or two at the initial learning phase will interrupt the learning process. Finally, the degree of learning-associated motor control could also be determined at the IO activity level. Behavioral learning may weaken CF activity over time, thereby decreasing the PC output onto the DCN, which then produces stronger motor output signals (*Low et al., 2018*).

The CF burst depends on the oscillatory state of the IO neurons, which are electrically coupled by gap junctions (*Lampl and Yarom, 1993*; *Mathy et al., 2009*). This oscillatory property is heterogeneous (*Hoge et al., 2011*), and each olivary neuron has a distinctive and stable oscillatory property (*Khosrovani et al., 2007*). Interestingly, the cerebellar cortex is compartmentalized in terms of Zebrin II expression in the PC, and the CS pattern also differs by zones (*Cerminara et al., 2015*). We observed that PC and CF $Ca^{2+}$ properties differ significantly by zones (*Roh et al., 2017*). Thus, we suspect that the heterogeneous property of oscillation of IO neurons, which determines CF burst, may shape the specific patterns of CSs across cortical zones, facilitating diversified control over CSs, and dendritic $Ca^{2+}$ transients in the PC.

This study suggests that CF pre-synaptic activity conveys variable and graded sensory information to post-synaptic PC in vivo, which is in line with other recent studies that denied the long-held 'all-or-none' notion for the CF activity (*Najafi et al., 2014b*; *Yang and Lisberger, 2014*; *Gaffield et al., 2019*). Such graded signal transmission is governed by the state of olivary oscillation—the number of spikes in the burst. Hence, it calls for the investigation of how IO sophisticatedly controls PCs over the whole cerebellum, which will unveil crucial mechanisms for cerebellar learning.

## Materials and methods

### Key resources table

| Reagent type (species) or resource | Designation | Source or reference | Identifiers | Additional information |
|---|---|---|---|---|
| Strain, strain background *Mus musculus* | B6.129-Tg(*Pcp2*-cre)2Mpin/J | Jackson Laboratory | RRID:IMSR_JAX:004146 | Stock no: 004146 |
| Strain, strain background *M.* | B6.Cg-Tg(*Camk2a*-cre)T29-1Stl/J | Jackson Laboratory | RRID:IMSR_JAX:005359 | Stock No: 005359 |

*Continued on next page*

*Continued*

| Reagent type (species) or resource | Designation | Source or reference | Identifiers | Additional information |
|---|---|---|---|---|
| Other | AAV1.*Camk2a.*GCaMP6f.WPRE.SV40 | Upenn Vector Core | | |
| Other | AAV1.CAG.FLEX.jRGECO1a.WPRE.SV40 | Upenn Vector Core | | |
| Other | AAV1.CAG.FLEX.GFP.WPRE.SV40 | Upenn Vector Core | | |
| Chemical compound, drug | Zoletil | Virvac | | |
| Chemical coumpound, drug | Rompun | Bayer | | |
| Chemical compound, drug | Dexamethasone | Samyang Phamaceutical | | |
| Chemical compound, drug | Meloxicam | Boehringer Ingelheim | | |
| Software, algorithm | MATLAB | Mathwroks Inc | RRID:SCR_002881 | |
| Software, algorithm | Python | https://www.python.org/ | RRID:SCR_008394 | |

## Animals, craniotomy, and genetically encoded Ca²⁺ indicator (GECI)

The experimental processes were approved by the Seoul National University Institutional Animal Care and Use Committee and performed under the guidelines of the National Institutes of Health. Seven- to ten-week-old wild-type or B6.129-Tg(*Pcp2*-cre)2Mpin/J (Jackson Laboratory, ME, USA) mice were anesthetized with intraperitoneal injections of Zoletil/Rompun mixture (30 mg / 10 mg/kg). A small craniotomy was made over lobule IV/V of the cerebellar vermis/paravermis according to previous descriptions but with some modifications (*Kim et al., 2016*). In short, after placing the anesthetized mouse on a stereotaxic frame (Narishige, Tokyo, Japan), the skin was incised, and bone was removed with a no.11 surgical blade. To minimize edema and related inflammation, dexamethasone (0.2 mg/kg) and meloxicam (20 mg/kg) were administered by subcutaneous injection. A metal ring for head fixation was attached with Superbond dental cement (Sun Medical, Japan). For PC-specific GECI expression, 100–200 nl virus solution of 3–5 × 10⁹ genome copies containing AAV1.CAG.FLEX.jRGECO1a.WPRE.SV40 (Upenn Vector Core, PA, USA) were injected at two or three sites at the cerebellar cortex of the *Pcp2-cre* TG mice with a beveled glass pipette (5 MΩ). Then, a 1.3 × 2.3 mm size glass coverslip (Matsunami, Japan) was tightly placed on the cortex and fixed by applying cyanoacrylate glue (Vetbond, 3M). For GCaMP6f expression in CF, the virus was injected into the IO 3–4 days before the creation of a chronic window, as previously described (*Kimpo et al., 2014*). Briefly, bilateral injections were made at the midpoint between the edge of the occipital bone and the C1 cervical vertebra. The glass pipette was set at a 55° angle from vertical and 7° from the midline. After approaching a 2.5 mm depth, virus solution containing 100–200 nl of AAV1.*Camk2a.*GCaMP6f.WPRE.SV40 was injected with a Picopump at 5 nl / sec. The pipettes were left in place for 10 min before they were removed to minimize backflowing. For GFP expression in CFs or PCs, AAV1.CAG.FLEX.GFP.WPRE.SV40 was injected into *Pcp2*-cre or *Camk2a*-cre mouse (B6.Cg-Tg(Camk2a-cre)T29-1Stl/J) (Jackson Laboratory, ME, USA).

## Two-photon microscopy and chronic awake imaging

Confocal microscopy was performed with a laser scanning multiphoton microscope (Zeiss LSM 7 MP, Carl Zeiss, Jena, Germany) equipped with non-descanned detectors (NDD). Excitation was carried out with a mode-locked titanium:sapphire laser system (Chameleon, Coherent, Santa Clara, CA, USA) operating at a 900 nm wavelength for GCaMP6f using GFP filter and 1030 nm for jRGECO1a using RFP filter. Generally, objective W Plan-Apochromat 20×, 1.0 numerical aperture (Carl Zeiss) was used. Images were acquired using ZEN software (Zeiss Efficient Navigation, Carl Zeiss) and processed using a custom-written MATLAB (MathWorks) script. High-resolution of 512 × 512 pixel

reference images were acquired at a rate of 8 s per frame in the PC layer (120–150 μm from the dura) and molecular layer (around 20–60 μm from the dura). For the 3D Z-stack images of the CF, 512 × 512 pixel images were acquired at every 2 μm from the dura to a 180 μm depth using the depth correction mode by ZEN. For PC and CF Ca²⁺ imaging, 32 Hz high-speed time-lapse scanning was performed with a 512 × 64 resolution window at 30–50 μm from the dura. For awake imaging, 10 days after chronic window surgery, the mice were subjected to handling until they show grooming (5–10 min) as well as acclimation on a custom-made disk treadmill with their head fixed using a clamp and custom-made metal rings (30–60 min) for 3 days. Imaging was performed 14 days after surgery.

## CF varicosity and PC dendritic Ca²⁺ signal analysis

CF Ca²⁺ imaging data were processed and analyzed with the open-source analysis tool Suite2p which efficiently detects Ca²⁺ signals both at individual axonal varicocities (*Marius Pachitariu et al., 2016*). After motion correction and source extraction by Suite2p, the ROIs with low signal-to-noise ratio were discarded and ones with proper size/morphology were selected for analysis in the Suite2p's graphic user interface (*Figure 1c*). If ROIs are located in the similar parasagittal plane (~20 μm) and their correlation coefficient of signals are above 0.85, the pairs were merged in Suite2p. Then fluorescence signals (F) were subtracted with neurophil (background) signals and the signals were z-score normalized for further analysis (https://github.com/NeuRoh1/Calcium_signal_processing; *Roh, 2020*; copy archived at swh:1:rev:b0732bd900ccf0e198f52818366f93715a49cea6). The event detection was performed using Suite2p's built-in deconvolution (tau: 0.4, window for maximum: 60, smoothing constant for gaussian filter: 25), and the deconvolved signals were scaled to match the amplitude of the z-scored F signals (*Figure 1d*). The amplitudes of CF Ca²⁺ transients were obtained from peak values of detected events. To ensure the event detection quality, the events with amplitudes lower than 0.5 SD were discarded. To analyze the synchrony of the Ca²⁺ spikes, Pearson's correlation coefficients for the firing patterns between every pair of ROIs were computed (*Figure 1e*). For 'synchrony by mediolateral distance' analysis, the mediolateral distance between pairs of all ROIs was obtained with ImageJ and paired with the corresponding calculated R values for correlation analysis (*Figure 1f*). PC Ca²⁺ signals imaged with jRGECO1a were similarly processed with Suite2p. The ROIs of PC dendrites were selected based on their structural identity that appears elongated along anterior-posterior axis (*Kitamura and Häusser, 2011*; *Ozden et al., 2009*; *Schultz et al., 2009*) and signal quality. Over-segmented ROIs were merged in Suite2p. For all analyses of CF and PC Ca²⁺ signals, we only included resting-state data and excluded the images during locomotion as we are interested in sensory processing, and it generates many complex Ca²⁺ transients.

## Two-photon dual-color Ca²⁺ imaging and analysis

For simultaneous imaging of the CF and PC, AAV1.*Camk2*.GCaMP6f.WPRE.SV40 was first bilaterally injected into the IO; after 1–3 days, AAV1.CAG.FLEX.jRGECO1a.WPRE.SV40 was then injected into the cerebellar cortex in *Pcp2-cre* TG mice while creating a chronic imaging window. The cerebellar cortex was excited with a 1000 nm wavelength, and the signals for GFP and RFP filters were simultaneously acquired. The CF varicosity and the PC dendrite were regarded as a pair when they spatially overlap and their signals are significantly similar (*Figure 4b,c*). To examine the relationship between CF and PC, we performed correlation analysis of the signals and of paired amplitudes. For the signal-correlation analysis, we computed the correlation coefficient between signals of each pair with MATLAB. If PC dendrites were spatially overlapped with CF, they were considered 'paired' CF-PC. The CF and the PC dendrites that were not overlapped were regarded as 'unpaired' CF-PC, which were further categorized as 'neighboring unpaired' if their boarders meet within 30 μm and 'distant unpaired' if the borders were separated by more than 30 μm. The signal correlations were compared in the four conditions (*Figure 4d*). For the amplitude-correlation analysis, the paired CF and PC amplitudes were collected where CF Ca²⁺ events occurred. All paired amplitudes from 1 min recordings were subjected to a non-linear fit analysis using GraphPad Prism (GraphPad Software Inc, CA, USA), and R² values and slope of the fit was presented (*Figure 4e–g*).

## Air-puff sensory stimulation, motion tracking, and correlation with Ca$^{2+}$

Animal sensory stimulation and motion tracking were controlled by a custom-written program in LabVIEW (National Instruments, USA). Briefly, the sensory stimulation and motion tracking were synchronized with two-photon imaging through a trigger generated by the program. Periocular air puffs lasting 30 ms were delivered with a Pneumatic Picopump (WPI, USA) at 5 s intervals, at 20 (P1) or 50 (P2) psi. Animal motion was acquired at 64 Hz using a high-speed CCD camera (IPX-VGA210, IMPERX, USA) under infrared (IR) illumination (DR4-56R-IR85, LVS, S. Korea). Motion tracking was achieved by tracking the IR-reflective patches (4 mm diameter) attached to the mice's forepaws. The velocity of the patch calculated on two dimensions (X and Y axes) was considered to be the animals' velocity of motion. Small forepaw movements induced by air-puff stimulation were successfully tracked (*Figure 3g*). To analyze the correlation between motion strength and CF Ca$^{2+}$ amplitude, the peak speeds were acquired within 200 ms after the onset of spontaneous CF Ca$^{2+}$ signal (for correlation during rest) or air-puff stimulation (for correlation during air puffs) (*Figure 3f–h*). The peak motion speed and CF Ca$^{2+}$ amplitudes were subjected to linear regression analysis using GraphPad Prism (GraphPad Software Inc, CA, USA), and R$^2$ values were presented.

## Ex vivo slice electrophysiology and Ca$^{2+}$ imaging

Acute preparation and an electrophysiological experiment were carried out as previously described (*Ryu et al., 2017*). Briefly, 5- to 9-week-old mice were anesthetized by isoflurane and decapitated. Then, 250 μm thick sagittal slices of the cerebellar vermis were obtained from mice using a vibratome (VT1200, Leica). The ice-cold cutting solution contained 75 mM sucrose, 75 mM NaCl, 2.5 mM KCl, 7 mM MgCl$_2$, 0.5 mM CaCl$_2$, 1.25 mM NaH$_2$PO$_4$, 26 mM NaHCO$_3$, and 25 mM glucose with bubbled 95% O$_2$ and 5% CO$_2$. The slices were immediately moved to artificial cerebrospinal fluid (ACSF) containing 125 mM NaCl, 2.5 mM KCl, 1 mM MgCl$_2$, 2 mM CaCl$_2$, 1.25 mM NaH$_2$PO$_4$, 26 mM NaHCO$_3$, and 10 mM glucose with bubbled 95% O$_2$ and 5% CO$_2$. Then, they were recovered at 32°C for 30 min and at room temperature for 1 hr. All of the recordings were performed within 8 hr of recovery.

The brain slices were placed in a submerged chamber with perfusion of ACSF for at least 10 min before recording. Whole-cell recordings were made at 29.5–30°C. We used recording pipettes (3–4 MΩ) filled with (in mM): 9 KCl, 10 KOH, 120 K-gluconate, 3.48 MgCl$_2$, 10 HEPES, 4 NaCl, 4 Na$_2$ATP, 0.4 Na$_3$GTP, and 17.5 sucrose (pH 7.25). Electrophysiological data were acquired using an EPC9 patch-clamp amplifier (HEKA Elektronik) and PatchMaster software (HEKA Elektronik) with a sampling frequency of 20 kHz, and the signals were filtered at 2 kHz. All of the electrophysiological recordings were acquired in lobule III–V of the cerebellar central vermis. The CFs were stimulated by ACSF-containing glass pipettes placed onto the granule cell layer.

For CF Ca$^{2+}$ imaging, slices were prepared from mice that had AAV1.*Camk2a*.GCaMP6f.WPRE. SV40 injected into their IO 3–4 weeks before the preparation. For PC Ca$^{2+}$ imaging with CS recording, the low-affinity Ca$^{2+}$ dye Fluo-5F (0.5 mM, F14221, Molecular Probes) was loaded from recording pipettes into the PC. A microscope (BX50W, Olympus) was equipped with a 40X objective lens (LUMPlanFLN, Olympus). Images were acquired at 5 Hz with a scientific CMOS camera (Prime, Photometrics). All of the experiments were performed in duplicate or triplicate, and randomly selected traces of each cell were analyzed.

## Statistics

Graph plotting and statistical analysis were carried out with GraphPad Prism (GraphPad Software Inc, CA). The hypothesis was tested by one-way ANOVA followed by post hoc Tukey's test, for multiple comparisons using either GraphPad Prism or MATLAB. Unpaired *t*-tests between sample pairs were carried out. The correlation between Ca$^{2+}$ signals and motion was analyzed by linear regression using GraphPad Prism. Results were considered significant if the P-value was below 0.05. Asterisks denoted in the graph indicate the statistical significance. * means p-value<0.05, **<0.01, ***<0.001, and ****<0.0001. The test name and statistical values are presented in each figure legend.

## Acknowledgements

We thank Jeong-Kyu Han, Gee-Hoon Chung, Suk-Chan Lee, and Yong Seok Lee for their helpful discussions as well as Dong-Cheol Jang for the graphical illustrations. This work was supported by National Research Foundation of Korea (NRF) grants funded by the South Korean government (2018R1A5A2025964 and 2017M3C7A1029611 to SJ Kim; NRF-2016R1D1A1A02937329 to SK Kim).

## Additional information

### Funding

| Funder | Grant reference number | Author |
|---|---|---|
| National Research Foundation of Korea | 2018R1A5A2025964 | Sang Jeong Kim |
| National Research Foundation of Korea | 2017M3C7A1029611 | Sang Jeong Kim |
| National Research Foundation of Korea | 2016R1D1A1A02937329 | Sun Kwang Kim |

The funders had no role in study design, data collection and interpretation, or the decision to submit the work for publication.

### Author contributions

Seung-Eon Roh, Conceptualization, Resources, Data curation, Software, Formal analysis, Funding acquisition, Validation, Investigation, Visualization, Methodology, Writing - original draft, Writing - review and editing; Seung Ha Kim, Data curation, Investigation; Changhyeon Ryu, Data curation, Validation, Investigation; Chang-Eop Kim, Resources, Data curation, Software, Methodology; Yong Gyu Kim, Resources, Software, Methodology; Paul F Worley, Supervision, Writing - review and editing; Sun Kwang Kim, Sang Jeong Kim, Conceptualization, Supervision, Funding acquisition, Writing - review and editing

### Author ORCIDs

Seung-Eon Roh (iD) https://orcid.org/0000-0002-1222-4365
Changhyeon Ryu (iD) https://orcid.org/0000-0001-5207-9142
Paul F Worley (iD) http://orcid.org/0000-0002-5086-614X
Sun Kwang Kim (iD) https://orcid.org/0000-0002-2649-6652
Sang Jeong Kim (iD) https://orcid.org/0000-0001-8931-3713

### Ethics

Animal experimentation: This study was performed in strict accordance with the recommendations in the Guide for the Care and Use of Laboratory Animals of the National Institutes of Health. All of the animals were handled according to approved institutional animal care and use committee (IACUC) protocols (#SNU-111214-6-3) of the Seoul National University. The protocol was approved by the Committee on the Ethics of Animal Experiments of the Seoul National Universtiy. All surgery was performed under intraperitoneal injections of Zoletil/Rompun mixture (30 mg / 10 mg/kg), and every effort was made to minimize suffering.

### Decision letter and Author response

Decision letter https://doi.org/10.7554/eLife.61593.sa1
Author response https://doi.org/10.7554/eLife.61593.sa2

## Additional files

### Supplementary files

- Transparent reporting form

### Data availability

All data generated or analysed during this study are included in the manuscript and supporting files. Source data files have been provided for Figures 1 through 5. Code has been made available via GitHub at https://github.com/NeuRoh1/Calcium_signal_processing (copy archived at https://archive.softwareheritage.org/swh:1:rev:b0732bd900ccf0e198f52818366f93715a49cea6).

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
