## [Decision Letter]

**Acceptance summary:**

In this study, Roh et al. investigate graded signaling between climbing fibers and cerebellar Purkinje cells. While classical theories of cerebellar learning treat climbing fiber input as an all-or-none binary signal, recent work has questioned this doctrine by demonstrating graded climbing fiber responses recorded postsynaptically at Purkinje cells. Such graded responses would change current thinking about how climbing fibers drive both synaptic plasticity and motor learning. Currently it remains unresolved how such graded responses are generated. Here the authors present evidence that the number of spikes in high frequency burst firing of olivary cells could be the main determinant of postsynaptic calcium influx amplitude in Purkinje cells. The authors perform simultaneous calcium recordings of CF varicosities and of their postsynaptic Purkinje cells in vivo during behavior and describe a strong correlation in amplitude between presynaptic and postsynaptic signals, suggesting that presynaptic burst size may control the postsynaptic response. Further, as has been reported for postsynaptic PC responses, the authors report that climbing fiber inputs represent levels of sensory stimulation in a graded manner.

**Decision letter after peer review:**

Thank you for submitting your article "Direct translation of climbing fiber burst-mediated sensory coding into post-synaptic purkinje cell dendritic calcium" for consideration by *eLife*. Your article has been reviewed by three peer reviewers, and the evaluation has been overseen by a Reviewing Editor and Ronald Calabrese as the Senior Editor. The reviewers have opted to remain anonymous.

The reviewers have discussed the reviews with one another and the Reviewing Editor has drafted this decision to help you prepare a revised submission.

Summary:

In this study, Roh et al. investigate whether and how cerebellar climbing fibers can encode sensory input in a graded manner. This question is crucial to understanding the mechanisms of cerebellar learning, because climbing fibers produce an instructional signal that can drive both heterosynaptic plasticity at cerebellar Purkinje cell synapses as well as some forms of motor learning. While classical theories of cerebellar learning treat climbing fiber input as an all-or-none binary signal, recent work has questioned this doctrine by demonstrating graded climbing fiber responses recorded postsynaptically at Purkinje cells, and currently it remains unresolved how such graded responses are generated.

Here the authors propose that the number of spikes in high frequency burst firing of olivary cells could be the main determinant of postsynaptic calcium influx amplitude in Purkinje cells. If demonstrated, this would constitute a new and exciting result with deep consequence on our understanding of cerebellar supervised learning. The authors perform simultaneous calcium recordings of CF varicosities and of their postsynaptic Purkinje cells in vivo during behavior and describe a strong correlation in amplitude between presynaptic and postsynaptic signals, suggesting that presynaptic burst size may control the postsynaptic response.

The authors show nicely that, as has been reported for postsynaptic PC responses, climbing fiber inputs represent levels of sensory stimulation in a graded manner. The major conclusion of this study is that the variability in Purkinje cell dendritic calcium signal amplitude and magnitude/duration of complex spikes are a direct result of variability in the size of climbing fiber input in vivo during sensory processing. While none of the data presented in this paper contradict this claim, the data presented are all either partial or circumstantial in support of this conclusion. Moreover, there are substantial technical concerns regarding the imaging analysis and data interpretation. If the authors can address these concerns to convincingly demonstrate that the CF Ca^2+^ transients are governed by the number of spikes in CF, and that these signals truly co-vary with the postsynaptic response, the reviewers believe the paper would be of significant interest.

Essential revisions:

1) The quantification of calcium imaging data

a) The legend of Figure 1 states that "independent ROIs" were selected. It is unclear how ROI selection was made for CF boutons – was this manual? And how were ROI's clustered into independent CFs, i.e., how was independence determined? A single climbing fiber makes many varicosities and innervates multiple Purkinje cells. This figure quantifies both climbing fiber firing frequency and synchrony, measurements that rely on measurements from unique climbing fibers. A simple spatial segregation of ROIs may not be sufficient to establish such independence. Observation of Figure 1C-E suggests that several of the ROIs actually correspond to the same climbing fiber, given their near perfect correlation. In particular, it looks like some ROIs that were analyzed separately may correspond to a single CF e.g. ROIs 2-3 and 4-5 in Figure 1. Related to this point, it is unclear why the correlation matrix heatmap in Figure 1E is smoothed. This is both confusing and misleading.

b) They state "Hence the results indicate that the variability of CF Ca^2+^ activity arises from its differential spike number of burst". I assume that the authors mean by this that the variability in CF Ca^2+^ amplitude is mainly driven by the number of spikes in the olivary response underlying it. Figure 1G is where they show the distribution of CF Ca^2+^ amplitudes in vivo. One presumes that this plot is obtained by aggregating all the data from the 60 cells they recorded. Therefore it stands to reason that a lot of the variability in the amplitudes is actually driven by cell to cell variation rather than the number of spikes in the burst. Indeed, this suspicion is supported by Figure 2D, where it is obvious that CF calcium amplitudes vary widely from cell to cell for a fixed number of spikes in the burst. To strengthen their conclusions, they should analyse their in-vivo data differently: rather than lumping all the Ca^2+^ responses from all the cells together, have they tried plotting amplitude histograms for each individual recording? If they could show separate peaks in such a histogram, that would represent quite convincing evidence that the amplitude of their CF Ca^2+^ responses are being driven by the number of spikes in the olivary burst. They could consider getting a population histogram by normalizing each cell's data to the first peak in its histogram.

c) The authors report correlation coefficient values for CF boutons in Figure 1H, but it is unclear over what spatial extent these were calculated. It would be more useful to report these as a function of mediolateral distance. An important, and easily testable prediction would be that these correlations should fall off at a similar spatial scale as Purkinje cell dendritic calcium signals (as in Ozden et al., J. Neuroscience, 2009, Figure 1D or Gaffield et al., J. Neurophysiology, 2016, Figure 4B). This sort of analysis may also provide a useful metric for defining CF ROIs that correspond to the same olivary neuron, as these are likely to exhibit higher correlations that may be represented as outliers to distribution that decays smoothly as a function of space (as observed for Purkinje cell calcium imaging).

d) There are now a variety of freely available software suites, such as CaImAn (https://elifesciences.org/articles/38173) or Suite2p (https://www.biorxiv.org/content/10.1101/061507v2), that allow for rigorous and standardized selection, curation, and analysis of imaging data. It may be advisable to use such tools for data analysis.

e) There are also no examples of how event detection was performed on the imaging data. It would be useful to show example traces with detected events. This is particularly important in Figure 4, where correspondence between CF imaging and PC imaging is necessary. Similar to the point above about image segmentation, there are also a variety of freely-available software suites that allow for event detection. It may be useful to use these. The reported methods state that signals were detected using a single threshold, but event detection algorithms take indicator kinetics into account to detect events with a particular shape (i.e. fast rise, slow decay) that match the likely waveforms of real events. In particular, I would recommend MLspike, which is well-suited for detecting sparse events that may overlap in time (https://github.com/MLspike/spikes).

2) Throughout the manuscript, there is a systematic lack of reporting of how many animals were used, how many repetitions were performed for different sets of experiments, and how group statistics were calculated:

a) Figure 1F-H: please report how many ROIs, how many fields of view, and how many mice were used in these analyses (and if any ROIs were resampled. The only mention of the number of samples here is regarding Figure 1H, and it states that the data come from 10 recordings (without mentioning the number of ROIs per recording, the number of recordings per animal, or the number of animals).

b) Figure 2: the data appear to be pooled from 9 CF imaged in 3 mice and the mean(?) response for each CF is shown as a thin line in Figure 2C, while the mean response across CFs is shown as a thick red line. It is unclear how many repetitions were performed per imaged CF. Furthermore, it is unclear whether a single, small, round ROI (i.e. a single bouton) was chosen as an ROI for each CF or whether multiple boutons were averaged per CF. Finally, it appears that a statistical comparison was only made between the single stimulation group and the other conditions. Are the other groups (i.e. 3, 5, 7, and 9 stimulations) different from each other?

c) Figure 3F-H: it is unclear how many CF boutons were in to the reported number of events. Given that there are several “event data points” (y-axis) for each "speed data point” (x-axis), it appears that multiple CFs were imaged simultaneously. This should be made explicit.

d) Figure 4: it is unclear how many CF-PC pairs went into the 184 “events” that are reported in panel D. It is also not clear if CF or PC transients were used to define these “events.” The example Purkinje cell ROI in panel B is also inconsistent in size and shape with the criteria reported in the manuscript's methods. Analyzing these data carefully is crucial, since the main conceptual advance of the paper is to show that CF signal amplitudes co-vary systematically with postsynaptic PC signal amplitudes.

e) The main observation of the paper, the correlation in Figure 4D, raises many questions. First, only 184 events are plotted, corresponding to about 3 minutes of recording overall. This is a very small number of observations for 3 mice, as many varicosity/dendrite appositions should be visible in a single field of view. Most importantly, the number of independent pre/post pairs is not stated and all the data are pooled. Thus co-variation in signal intensity between pairs could be mistaken for co-variation within a pair. Indeed, Figure 2C shows a ten-fold variation in DF/F from one CF to the other, which should obscure the correlation in Figure 4D. Astonishingly, in vitro transients saturate at about 50 % DF/F (Figure 1C) for long bursts, while in vivo calcium transients in varicosities range from 80 % to 700 % (Figure 1D,G and 4C,D). All these issues should be addressed and in particular correlation should be established pair by pair, with many more events, not for the whole set of data.

f) Figure 5: There is no mention of the number of repetitions that were performed per cell here.

3) Small structures such as the varicosities imaged here can easily move into and out of the focal plane when there is subtle brain movement, producing artificially larger DF/F responses. There is no mention in the methods or figures of how motion correction was applied to bouton imaging data, which is likely to contain motion artifacts because the animals were awake and locomoting and the imaged structures are quite small. Was such correction done? If so, how? This is particularly striking, given how stable the baseline appears in Figures 1D, 3B-C If any transient movement (not necessarily the paw movement) is correlated with the intensity of stimulation, this could produce the “graded” responses in Figure 3. Movement artifacts could be assessed by imaging GFP in climbing fiber varicosities during stimulation, or by demonstrating that any other stimulus-induced movement is correlated in amplitude with paw movement. Related to this point, it is not clear to me what is plotted in Figure 3 F and H.

4) The contribution of movement to “sensory driven” responses is not sufficiently characterized.

a) The authors use IR tracking of a forepaw to reveal animal movement, but a periocular air puff is likely to produce orofacial movements not captured by paw tracking. The authors show no correlation between the amplitude of paw movements and peak calcium transients, but paw motion may not be the relevant movement that enhances climbing fiber responses in this recording location. The authors should demonstrate that the amplitude of paw movements correlates with the amplitude of any other movements generated by the air puff, or if that is not possible, at the very least, allow for this possible confound in their data.

b) If I understand Figure 3 G correctly, a transient paw movement is triggered by air puff stimulation. Is this distinct from the movement plotted in Figure 3 F and H, which appears to be a peak paw speed associated with running at the time of stimulation in H, and in absence of stimulation in F? Clearly in F, there is no stimulation, so paw movements must be related to running? In any case, it is not evident that running speed in panel F and the transient (unexpected) paw movement generated by puffs are the same. If not, there may be a correlation between the transient paw movement and climbing fiber activity independent of running.

5) The literature demonstrating the influence of postsynaptic factors on the CF calcium transient in Purkinje cells (Kitamura and Hausser 2011, Otsu et al. 2014, Gaffield et al. 2018 to cite only a few recent) is mostly overlooked. Similarly, presynaptic modulations able to change presynaptic calcium influx in CF varicosities and therefore glutamate release should be discussed, as a possible additional source of variability on top of the number of spikes in bursts.

There are several assertions made in the paper regarding the relationship between CF bursts (action potentials) and CF axonal calcium responses. While portions of the statements made by the authors are true, they sometimes apply faulty logic to arrive at their conclusions. Specifically:

a) “ Our data indicate that increasing the number of burst stimulation

significantly augments the CF Ca^2+^ amplitude, which is saturated at 7 bursts (Figure 2C,D).”: – While it is true (as the authors show) that increasing the stim number in a burst increases the CF Ca^2+^ signal amplitude, this does not definitively prove their next sentence “…the variability CF Ca^2+^ activity arises from its differential spike number of burst.” Just because the first statement is true, does not mean the second one (its converse) is also true.

b) Results related to Figure 5 – The title of this section is “The spike number of CF burst determines the amplitudes of PC dendritic Ca^2+^ response.” While their data show that these events are correlated, this statement is not uniquely true. For example, Wang et al. (Nature Neuro, 2000), Kitamura and Hausser (J. Neuroscience, 2011), and others have demonstrated that PC dendritic calcium signals are related to the interactions between PF inputs, CF inputs, and the intrinsic properties of PCs. Thus, the authors' statement is not uniquely true.

[Editors' note: further revisions were suggested prior to acceptance, as described below.]

Thank you for submitting your work entitled "Direct Translation of Climbing Fiber Burst-Mediated Sensory Coding into Post-Synaptic Purkinje Cell Dendritic Calcium" for consideration by *eLife*. Your article has been reviewed by two peer reviewers, and the evaluation has been overseen by a Reviewing Editor and a Senior Editor. The reviewers have opted to remain anonymous.

Our decision has been reached after consultation between the reviewers. Based on these discussions and the individual reviews below, we regret to inform you that your work will not be considered further for publication in *eLife*.

Although the reviewers were appreciative of your efforts to improve the manuscript with this revision, it was agreed in the consultation phase that the original technical concerns persist and have not been sufficiently overcome. In particular, the issues raised by reviewer 1 about ROI detection, quantification of event amplitudes, and the identification of CF-PC pairs are seen as critical and were not perceived to be adequately defended in the manuscript. Given these concerns, together with the lack of significant differences between the unpaired neighbors and the paired neighbors, the reviewers do not believe that the conclusion that climbing fiber (CF) high frequency bursts are responsible for the graded amplitude of the postsynaptic calcium transients in the Purkinje cells (PCs) is adequately supported by the data.

Reviewer #1:

In the revised version of their paper, the authors have performed an extensive new analysis of their dataset using state of the art methods. Data are quantified and displayed in a more detailed way and the paper is overall substantially improved. However, the results still fail to support convincingly the main conclusion of the paper, namely that climbing fiber (CF) high frequency bursts are responsible for the graded amplitude of the postsynaptic calcium transients in the Purkinje cells (PCs). The identification of individual CFs is problematic, and equating CF calcium transient amplitude with CF firing bursts remains correlative. Furthermore, the claim that CF-PC pairs are recorded does not seem warranted based on the data provided. In conclusion the correlations between CF and PC may result from other factors than the proposed burst coding.

1) The main issue still arises from the identification of CF ROIs and CF-PC pairs, a problem raised in the first round of reviews. The additional data provided in the revised version strongly suggest that CF ROIs defined by the authors encompass multiple CFs.

a) Morphological arguments

All data from the literature that traced the projections of single olivary cells point to the fact that the target Purkinje cells are not clustered at a given antero-posterior position but randomly spread in a thin parasagittal band. Therefore, each climbing fiber should only encompass a few varicosities in strict parasagittal alignment within a single PC dendrite, much thinner than the average 30 µm extension described in the paper.

b) Functional arguments

Functional studies on PC calcium transients indicate that PC cells displaying calcium transients with close to 100 % synchrony (hence receiving the same CF) are the exception rather than the rule (Ozden et al. 2009). Careful observation of the movies in the present dataset indicates that the only varicosities that display 100 % synchrony in their calcium transients are indeed strictly in parasagital alignment, hence contacting a single Purkinje cell.

c) In conclusion, the authors should detect events on single varicosities, which are easily identified manually and then check for 100% correlation to group varicosities in a single CF. It is hard to understand why the automatic algorithm defines such contorted ROIs covering large regions with no detectable signal. High level of correlation between nearby CF and poor signal to noise levels (Figure 1—figure supplement 1B) may explain the pooling of multiple CFs in a single ROI.

2) The improper identification of single climbing fibers affects the interpretation of the supposed paired recordings of Figure 4. CF1 in Figure 4B and D seems to be two fibers, one contacting PC1 and the other PC3. However, correlation with PC2, on which there is no contact appears as good, probably because PC1 and PC2 are themselves highly correlated. More intriguing, some calcium events which are obvious in PCs 1-3 of the color raster of Figure 4D do not show in the CF1. This may indicate that this(ese) CF do not actually contact these PCs or that detection of CF events is improper. All calcium transients seen in the PC should be found in the CF and conversely.

Panels 4G-H provide a much improved analysis of the paired data, based on individual CF-PC pairs. However, considering that CF ROIs cover multiple CFs, they can be interpreted in terms of CF population synchrony. Larger CF transients would correspond to multiple CF being active at the same time, which would in turn translate in larger PC transients, as the link between population synchrony and PC calcium transients has already been described by Najafi et al. (2014). This does not preclude burst coding to be participating to the graded encoding of synchrony, as larger errors may both stimulate more olivary cells and trigger bursts, but the data do not convincingly demonstrate this point. In particular the CF-PC correlation is not significantly different between paired and neighbor unpaired, in line with the interpretation that correlation results from populational synchrony at short distance and not from direct pairing.

3) Quantification of the transients is also a problem. It is entirely unclear how events are detected after non-negative deconvolution and denoising (which may be subject to caution given the signal to noise of the recordings and the possibility of overfitting, see Figure 1—figure supplement 1B). Failure to detect close-by events may arise in the report of artificially large event amplitudes. When multiple events are detected does the amplitude correspond to the summed peak of the DF/F trace or to the actual individual events amplitudes?

The very large amplitude of the CF transients (more than 2000 % in Figure 1) is surprising, as it may exceed the dynamic range of GCaMP6, particularly considering that ROIs cover large regions without any signal. Baseline subtraction, which may explain these large numbers if excess subtraction is applied, appears to be a recurrent issue in the paper with some panels displaying very low background noise (RFP in Figure 4 supplement 1a) and other very high (GFP in the same figure and in panel 1B).

Furthermore, the signals appear saturated in many of the figures as well as in the movies, which does not really allow to judge the dynamic range and shape of the activated elements. The two movies have compression issues which do not allow clear visualization on a frame by frame mode.

Finally, data of Figure 4 and the two movies originate from the same field of view and the two movies represent the same 20s but with very different levels of background subtraction and saturation. It is hard to understand why the same data are presented twice, and why not provide a more extensive sample of the data. Is it on purpose? If so, it should be explicitly mentioned.

Reviewer #2:

Roh et al. have revised their manuscript to sufficiently account for my previous concerns, most importantly by re-analyzing all of their calcium imaging data and enhancing the transparency of their data and methodological reporting.

Notably, I still have some concerns over the role of movement in the amplitude of the reported signals. The authors correlate DF/F with paw speed, but not the amplitude of paw movement, or any other movements the animals make in response to airpuff stimulation. However, they now perform additional controls suggesting that imaging artifacts from bouton movement do not contribute to their results, and they have tempered their conclusions regarding the role of movement appropriately. Given the challenging nature of this problem, the level with which the authors have now addressed the role of movement seems reasonable. Likewise, the enhanced analysis of “independent” CF ROIs remains less than airtight, but in my opinion now meets a reasonable standard.

Finally, there remain some typos, and some odd writing choices. Examples include statements like "this notion has been prevailed", and "sensory inputs disappear ex vivo". It is also atypical to end a manuscript with a final sentence telling the reader what the lab is working on next, rather than an overall statement about the significance of this study.

Overall, however, the manuscript now convincingly makes an important link between the presynaptic activity of climbing fibers and the postsynaptic calcium transient in Purkinje cells in awake animals. It thus provides a significant advance, in parallel with recent work from the Christie lab, to our understanding of cerebellar learning mechanisms.

---

## [Author Response]

Essential revisions:1) The quantification of calcium imaging data

We agree with the reviewers’ point that our data analysis and interpretation should maintain objectivity in terms of selecting ROIs. Here we have utilized CaImAn, the open source software tool to detect single CFs (Giovannucci et al., 2019). It includes non-rigid motion correction, constrained nonnegative matrix factorization (CNMF) algorithm for source extraction and sparse non-negative deconvolution for event detection.

a) The legend of Figure 1 states that "independent ROIs" were selected. It is unclear how ROI selection was made for CF boutons – was this manual? And how were ROI's clustered into independent CFs, i.e., how was independence determined? A single climbing fiber makes many varicosities and innervates multiple Purkinje cells. This figure quantifies both climbing fiber firing frequency and synchrony, measurements that rely on measurements from unique climbing fibers. A simple spatial segregation of ROIs may not be sufficient to establish such independence. Observation of Figure 1C-E suggests that several of the ROIs actually correspond to the same climbing fiber, given their near perfect correlation. In particular, it looks like some ROIs that were analyzed separately may correspond to a single CF e.g. ROIs 2-3 and 4-5 in Figure 1. Related to this point, it is unclear why the correlation matrix heatmap in Figure 1E is smoothed. This is both confusing and misleading.

As CaImAn was designed to be suitable for soma calcium signal detection, we slightly modified the analysis protocol by selecting the initialization method as “sparse NMF,” a method that is appropriate for detecting arbitrary structures such as axons rather than circular soma. Processing CF calcium imaging data with CaImAn gave many ROIs that are overlapping each other, so we performed manual curation of ROIs (4 out of 12 ROIs) and the ones overlapping or with bad signal were excluded by a human expert. This process was depicted in Figure 1—figure supplement 1. Independent CFs were well identified as axonal arborization structures that have mediolateral size of about 30.2 µm in average (Figure 1C, D, G) and their synchrony are moderately reduced with mediolateral separation (Figure 1E, F). Thus, we have successfully detected independent CFs and their calcium signals using the modified CaImAn.

b) They state "Hence the results indicate that the variability of CF Ca^2+^ activity arises from its differential spike number of burst". I assume that the authors mean by this that the variability in CF Ca^2+^ amplitude is mainly driven by the number of spikes in the olivary response underlying it. Figure 1G is where they show the distribution of CF Ca^2+^ amplitudes in vivo. One presumes that this plot is obtained by aggregating all the data from the 60 cells they recorded. Therefore it stands to reason that a lot of the variability in the amplitudes is actually driven by cell to cell variation rather than the number of spikes in the burst. Indeed, this suspicion is supported by Figure 2D, where it is obvious that CF calcium amplitudes vary widely from cell to cell for a fixed number of spikes in the burst. To strengthen their conclusions, they should analyse their in-vivo data differently: rather than lumping all the Ca^2+^ responses from all the cells together, have they tried plotting amplitude histograms for each individual recording? If they could show separate peaks in such a histogram, that would represent quite convincing evidence that the amplitude of their CF Ca^2+^ responses are being driven by the number of spikes in the olivary burst. They could consider getting a population histogram by normalizing each cell's data to the first peak in its histogram.

According to the reviewer’s suggestion, we have plotted amplitude histograms for each individual 3 minute recording for 11 cells (Figure 2D, E). As calcium spikes closely following the 1^st^ event will present an additive effect on amplitudes, only events within the first 0.5 seconds were used for analysis, as shown in Figure 2D. The amplitude histogram for the 11 cells shows a “discrete” distribution, suggesting several clusters of events with similar amplitudes (Figure 2E). These data would suggest that CF calcium responses are driven by the number of spikes in the olivary burst.

c) The authors report correlation coefficient values for CF boutons in Figure 1H, but it is unclear over what spatial extent these were calculated. It would be more useful to report these as a function of mediolateral distance. An important, and easily testable prediction would be that these correlations should fall off at a similar spatial scale as Purkinje cell dendritic calcium signals (as in Ozden et al., J. Neuroscience, 2009, Figure 1D or Gaffield et al., J. Neurophysiology, 2016, Figure 4B). This sort of analysis may also provide a useful metric for defining CF ROIs that correspond to the same olivary neuron, as these are likely to exhibit higher correlations that may be represented as outliers to distribution that decays smoothly as a function of space (as observed for Purkinje cell calcium imaging).

We have computed synchrony values between CFs in terms of mediolateral distance (Figure 1F). Similar to the previous results from the Purkinje cell calcium imaging, the correlation decreased as a function of distance. The identification of CF ROIs was carried out by clustering pixels that fire together using the CaImAn CNMF algorithm. The average mediolateral width of CF ROIs was 30.2 ± 2.9 (SEM) µm with the largest at 75.8 µm.

d) There are now a variety of freely available software suites, such as CaImAn (https://elifesciences.org/articles/38173) or Suite2p (https://www.biorxiv.org/content/10.1101/061507v2), that allow for rigorous and standardized selection, curation, and analysis of imaging data. It may be advisable to use such tools for data analysis.

As mentioned above, we have successfully adapted CaImAn to CF calcium signal analysis in combination with human expert curation for selecting ROIs out of a suggested ROI set. We have uploaded the code that runs CaImAn for CF Ca2^+^ signal analysis including the parameters as well as the code that performs curation to

GitHub repository (https://github.com/NeuRoh1/Calcium-analysis-tools).

e) There are also no examples of how event detection was performed on the imaging data. It would be useful to show example traces with detected events. This is particularly important in Figure 4, where correspondence between CF imaging and PC imaging is necessary. Similar to the point above about image segmentation, there are also a variety of freely-available software suites that allow for event detection. It may be useful to use these. The reported methods state that signals were detected using a single threshold, but event detection algorithms take indicator kinetics into account to detect events with a particular shape (i.e. fast rise, slow decay) that match the likely waveforms of real events. In particular, I would recommend MLspike, which is well-suited for detecting sparse events that may overlap in time (https://github.com/MLspike/spikes).

In this revision, we have utilized the non-negative deconvolution method (Vogelstein et al., 2010), which has been used for PC Ca^2+^ event detection (Gaffield et al., 2016), as well as CF and PC Ca^2+^ event detection (CaImAn also utilizes this algorithm). As shown in Figure 1—figure supplement 1b and Figure 3—figure supplement 2c, event detection was successful for each cell type. The event detection of CF-PC pairs from dual imaging also reveals a high degree of correspondence (Figure 4C).

As for PC Ca^2+^ analysis, we first performed non-rigid motion correction (NoRMCorre), and manually selected PC dendrites based on the structural uniformity and signal properties (human expert). The event detection was performed using sparse non-negative deconvolution algorithm (Vogelstein et al., 2010). For the source extraction, we’ve tried principle component analysis followed by independent component analysis (PCA/ICA) (Mukamel et al., 2009). However, the signal quality has no advantages against human expert analysis supported by NoRMCorre and deconvolution (Author response image 1). Also, as PCA/ICA processing misses some ROIs, we proceeded with human expert ROI selection (Figure 3—figure supplement 2).

**Author response image 1. sa2fig1:** 

2) Throughout the manuscript, there is a systematic lack of reporting of how many animals were used, how many repetitions were performed for different sets of experiments, and how group statistics were calculated:a) Figure 1F-H: please report how many ROIs, how many fields of view, and how many mice were used in these analyses (and if any ROIs were resampled. The only mention of the number of samples here is regarding Figure 1H, and it states that the data come from 10 recordings (without mentioning the number of ROIs per recording, the number of recordings per animal, or the number of animals).

The number of ROIs, FOV, and recordings used for analysis are now all clearly stated in the figure legend (Figure 1F,G).

b) Figure 2: the data appear to be pooled from 9 CF imaged in 3 mice and the mean(?) response for each CF is shown as a thin line in Figure 2C, while the mean response across CFs is shown as a thick red line. It is unclear how many repetitions were performed per imaged CF. Furthermore, it is unclear whether a single, small, round ROI (i.e. a single bouton) was chosen as an ROI for each CF or whether multiple boutons were averaged per CF. Finally, it appears that a statistical comparison was only made between the single stimulation group and the other conditions. Are the other groups (i.e. 3, 5, 7, and 9 stimulations) different from each other?

Five round CF axonal varicosities of < 2 µm that respond to stimuli were chosen as ROIs and then averaged for a single CF trace. 1 or 2 repetitions were performed and traces were randomly selected for analysis. This method is described in the Materials and methods section. In addition, all the significant comparisons are presented in the figure and figure legend (between 1 vs 5, 1 vs 7, 1 vs 9 and 3 vs 9 stim).

c) Figure 3F-H: it is unclear how many CF boutons were in to the reported number of events. Given that there are several “event data points” (y-axis) for each "speed data point” (x-axis), it appears that multiple CFs were imaged simultaneously. This should be made explicit.

We have reanalyzed the CF calcium imaging data with CaImAn and plotted the correlation graphs again. The number of events, recordings, animals are clearly presented in Figure 3 legend. The number of CFs is not reported since the number varies (3 – 6 CFs).

d) Figure 4: it is unclear how many CF-PC pairs went into the 184 “events” that are reported in panel D. It is also not clear if CF or PC transients were used to define these “events.” The example Purkinje cell ROI in panel B is also inconsistent in size and shape with the criteria reported in the manuscript's methods. Analyzing these data carefully is crucial, since the main conceptual advance of the paper is to show that CF signal amplitudes co-vary systematically with postsynaptic PC signal amplitudes.

We take this comment very seriously and as such performed an in-depth correlation analysis between CF and PC Ca^2+^ signals: this included signal correlation as well as amplitude correlation analysis. We first identified CF signals by CaImAn and selected PC dendrites within the established CF territories. These pairs are regarded as “paired” CF-PC. When the PC dendrites are out of the CF territories, they are regarded as “unpaired” CF-PC (Figure 4B-D). Such “unpaired” CF-PC were further categorized as “neighboring unpaired” if CFs’ borders are touching each other and “distant unpaired” if the borders are at least 30 µm apart. Here, we report highest signal correlation in paired CF-PC with reduced correlation depending on mediolateral distance (Figure 4C,D). For the amplitude correlation analysis, CF and PC amplitudes were used where CF Ca^2+^ transients take place. The number of events, pairs, recording session, and animal numbers are clearly stated in the figure legend (Figure 4).

e) The main observation of the paper, the correlation in Figure 4D, raises many questions. First, only 184 events are plotted, corresponding to about 3 minutes of recording overall. This is a very small number of observations for 3 mice, as many varicosity/dendrite appositions should be visible in a single field of view. Most importantly, the number of independent pre/post pairs is not stated and all the data are pooled. Thus co-variation in signal intensity between pairs could be mistaken for co-variation within a pair. Indeed, Figure 2C shows a ten-fold variation in DF/F from one CF to the other, which should obscure the correlation in Figure 4D. Astonishingly, in vitro transients saturate at about 50 % DF/F (Figure 1c) for long bursts, while in vivo calcium transients in varicosities range from 80 % to 700 % (Figure 1D,G and 4C,D). All these issues should be addressed and in particular correlation should be established pair by pair, with many more events, not for the whole set of data.

As the reviewer suggested, we have established the correlation pair by pair with a larger number of event pairs (1661 pairs) in this revision (Figure 4E-G). We provide pair by pair correlation for 19 CF-PC pairs (Figure 4G), while presenting the example of correlation of single pair (Figure 4E) and 1CF-3PC pairs (Figure 4F). Although there is a discrepancy in amplitude saturation between in vitro and in vivo transients, there are apparently many differences between them, such as imaging condition (widefield imaging vs 2-photon imaging), health of sample (sliced brain tissue vs intact brain), and baseline intensity (already increased in slice vs very low baseline). We also provide a video for the CF-PC dual imaging (Video 1).

f) Figure 5: There is no mention of the number of repetitions that were performed per cell here.

Now we state that the number of experiments is “7 recordings of 7 independent cells from 3 mice” (Figure 5). 1 or 2 repetitions were performed, but only the representative data were analyzed.

3) Small structures such as the varicosities imaged here can easily move into and out of the focal plane when there is subtle brain movement, producing artificially larger DF/F responses. There is no mention in the methods or figures of how motion correction was applied to bouton imaging data, which is likely to contain motion artifacts because the animals were awake and locomoting and the imaged structures are quite small. Was such correction done? If so, how? This is particularly striking, given how stable the baseline appears in Figures 1D, 3B-C If any transient movement (not necessarily the paw movement) is correlated with the intensity of stimulation, this could produce the “graded” responses in Figure 3. Movement artifacts could be assessed by imaging GFP in climbing fiber varicosities during stimulation, or by demonstrating that any other stimulus-induced movement is correlated in amplitude with paw movement. Related to this point, it is not clear to me what is plotted in Figure 3 F and H.

In this revision, we processed and analyzed all CF data with CaImAn, which includes motion correction by “non-rigid motion correction (NoRMCorre)” (Pnevmatikakis and Giovannucci, 2017). It gives a very stable video, which is uploaded as a video file for the CF-PC dual color imaging (Video 2). Moreover, CaImAn employs a template-matching algorithm to detect calcium events with fast rise and slow decay. Also, here we present 2-photon in vivo GFP imaging results under periocular air puff stimulation. These results show that air puff-induced movement is not deforming to CF-GFP and PC-GFP for both P1 and P2 stimulation (Figure 3—figure supplement 1). We also present a video showing before and after motion correction using NoRMCorre for both CF and PC during air puff stimulation (Video 2).

Previously we did not clearly describe that the motion we observed was not locomotion or walking/running but rather a subtle twitch-like movement of the forepaw induced by air puff stimulation. We apologize for any confusion the omission of this detail may have caused. We exclude the recordings where animals walk or run because we are interested in sensory coding at rest and locomotion produces a lot of complex Ca^2+^ signals. Figure 3F,H presents the correlation of CF calcium amplitude and forepaw twitch movement at rest. We claim that air puff stimulation-induced CF calcium transient is not related to the peak speed of air puff-induced twitch movement. These additional details and clarifications are now included in the Results and Materials and methods sections.

4) The contribution of movement to “sensory driven” responses is not sufficiently characterized.a) The authors use IR tracking of a forepaw to reveal animal movement, but a periocular air puff is likely to produce orofacial movements not captured by paw tracking. The authors show no correlation between the amplitude of paw movements and peak calcium transients, but paw motion may not be the relevant movement that enhances climbing fiber responses in this recording location. The authors should demonstrate that the amplitude of paw movements correlates with the amplitude of any other movements generated by the air puff, or if that is not possible, at the very least, allow for this possible confound in their data.

The movement we observed is a general movement for an unexpected stimulus. Although we have observed that eyeblink and forepaw movement occur simultaneously upon stimulation, we have now clearly stated the possibility of a confounding link to any other movements generated by air puff: “Although we did not examine any other movements related to air-puff stimulation, such as orofacial movement, we suggest that the CF Ca^2+^ activity could differentiate the graded sensory stimuli but not movement strength in an assumption that fore-paw motion is representing startle movement during air-puff stimulation.”

b) If I understand Figure 3G correctly, a transient paw movement is triggered by air puff stimulation. Is this distinct from the movement plotted in Figure 3F and H, which appears to be a peak paw speed associated with running at the time of stimulation in H, and in absence of stimulation in F? Clearly in F, there is no stimulation, so paw movements must be related to running? In any case, it is not evident that running speed in panel F and the transient (unexpected) paw movement generated by puffs are the same. If not, there may be a correlation between the transient paw movement and climbing fiber activity independent of running.

The confusion the reviewer indicated is because we previously did not state that we excluded the data with locomotion (see also our response to the above comment #3). The “spontaneous response” data in Figure 3F are limited to small and transient movements with CF Ca^2+^ events but not during walking or running. The peak speed is defined by the peak speed of fore-paw speed within 200 ms after the movement onset for spontaneous response (Figure 3F) or stimulation onset for air puff-induced movement (Figure 3H). These definitions are now described in the manuscript.

5) The literature demonstrating the influence of postsynaptic factors on the CF calcium transient in Purkinje cells (Kitamura and Hausser 2011, Otsu et al. 2014, Gaffield et al. 2018 to cite only a few recent) is mostly overlooked. Similarly, presynaptic modulations able to change presynaptic calcium influx in CF varicosities and therefore glutamate release should be discussed, as a possible additional source of variability on top of the number of spikes in bursts.There are several assertions made in the paper regarding the relationship between CF bursts (action potentials) and CF axonal calcium responses. While portions of the statements made by the authors are true, they sometimes apply faulty logic to arrive at their conclusions. Specifically:a) “ Our data indicate that increasing the number of burst stimulationsignificantly augments the CF Ca^2+^ amplitude, which is saturated at 7 bursts (Figure 2C,D).”: While it is true (as the authors show) that increasing the stim number in a burst increases the CF Ca^2+^ signal amplitude, this does not definitively prove their next sentence “…the variability CF Ca^2+^ activity arises from its differential spike number of burst.” Just because the first statement is true, does not mean the second one (its converse) is also true.

In this revision, we provide additional evidence that in vivo CF Ca^2+^ amplitude distribution appears discrete, which could further support the CF spike number-driven variability of CF Ca^2+^ amplitudes. However, we toned down the conclusion sentence as “Hence, the huge variability in CF Ca^2+^ amplitude may be formed by the variable number of CF burst.”

b) Results related to Figure 5 – The title of this section is “The spike number of CF burst determines the amplitudes of PC dendritic Ca^2+^ response.” While their data show that these events are correlated, this statement is not uniquely true. For example, Wang et al. (Nature Neuro, 2000), Kitamura and Hausser (J. Neuroscience, 2011), and others have demonstrated that PC dendritic calcium signals are related to the interactions between PF inputs, CF inputs, and the intrinsic properties of PCs. Thus, the authors' statement is not uniquely true.

According to the reviewer’s recommendation, the section title was changed to “The spike number of CF burst directly affects the amplitudes of PC dendritic Ca^2+^ response.” Also, the other contributions (intrinsic property of PC, PF activity, and noradrenergic CF control) to PC Ca^2+^ response were discussed in the Discussion section.

[Editors' note: further revisions were suggested prior to acceptance, as described below.]

Reviewer #1:In the revised version of their paper, the authors have performed an extensive new analysis of their dataset using state of the art methods. Data are quantified and displayed in a more detailed way and the paper is overall substantially improved. However, the results still fail to support convincingly the main conclusion of the paper, namely that climbing fiber (CF) high frequency bursts are responsible for the graded amplitude of the postsynaptic calcium transients in the Purkinje cells (PCs). The identification of individual CFs is problematic, and equating CF calcium transient amplitude with CF firing bursts remains correlative. Furthermore, the claim that CF-PC pairs are recorded does not seem warranted based on the data provided. In conclusion the correlations between CF and PC may result from other factors than the proposed burst coding.1) The main issue still arises from the identification of CF ROIs and CF-PC pairs, a problem raised in the first round of reviews. The additional data provided in the revised version strongly suggest that CF ROIs defined by the authors encompass multiple CFs.a) Morphological argumentsAll data from the literature that traced the projections of single olivary cells point to the fact that the target Purkinje cells are not clustered at a given antero-posterior position but randomly spread in a thin parasagittal band. Therefore, each climbing fiber should only encompass a few varicosities in strict parasagittal alignment within a single PC dendrite, much thinner than the average 30 µm extension described in the paper.b) Functional argumentsFunctional studies on PC calcium transients indicate that PC cells displaying calcium transients with close to 100 % synchrony (hence receiving the same CF) are the exception rather than the rule (Ozden et al. 2009). Careful observation of the movies in the present dataset indicates that the only varicosities that display 100 % synchrony in their calcium transients are indeed strictly in parasagital alignment, hence contacting a single Purkinje cell.c) In conclusion, the authors should detect events on single varicosities, which are easily identified manually and then check for 100% correlation to group varicosities in a single CF. It is hard to understand why the automatic algorithm defines such contorted ROIs covering large regions with no detectable signal. High level of correlation between nearby CF and poor signal to noise levels (Figure 1—figure supplement 1B) may explain the pooling of multiple CFs in a single ROI.

We admit that the identification of CF ROIs and consequently CF-PC pairs was problematic, compounding the interpretation of the results. Reviewing the literatures and data we conclude that CaImAn is not appropriate for CF Ca^2+^ analysis because it detects too large area (~30um) as indicated by the reviewer #1. Here, we tested and verified that another open source tool Suite2p effectively detects individual CF varicosities (Figure 1). If ROIs are located in the similar parasagittal plane (~20 um) and their correlation coefficient of signals are above 0.85, the pairs were merged in Suite2p GUI, resulting in less than 10 μm size of each ROI mediolaterally. Some CF ROIs appeared small which are less than 2-5 um, while some portion of bigger ROIs appeared parasagittaly aligned (i.e. ROI #1, 2, 7, 9, 13 in Figure 1C). Confirming that Suite2p analysis ensures proper interpretation of the results, we reanalyzed whole data set with the new method established and dispelled the concerns about CF ROI identification.

2) The improper identification of single climbing fibers affects the interpretation of the supposed paired recordings of Figure 4. CF1 in Figure 4B and D seems to be two fibers, one contacting PC1 and the other PC3. However, correlation with PC2, on which there is no contact appears as good, probably because PC1 and PC2 are themselves highly correlated. More intriguing, some calcium events which are obvious in PCs 1-3 of the color raster of Figure 4D do not show in the CF1. This may indicate that this(ese) CF do not actually contact these PCs or that detection of CF events is improper. All calcium transients seen in the PC should be found in the CF and conversely.Panels 4G-H provide a much improved analysis of the paired data, based on individual CF-PC pairs. However, considering that CF ROIs cover multiple CFs, they can be interpreted in terms of CF population synchrony. Larger CF transients would correspond to multiple CF being active at the same time, which would in turn translate in larger PC transients, as the link between population synchrony and PC calcium transients has already been described by Najafi et al. (2014). This does not preclude burst coding to be participating to the graded encoding of synchrony, as larger errors may both stimulate more olivary cells and trigger bursts, but the data do not convincingly demonstrate this point. In particular the CF-PC correlation is not significantly different between paired and neighbor unpaired, in line with the interpretation that correlation results from populational synchrony at short distance and not from direct pairing.

The data analysis and its interpretation related to simultaneous CF-PC Ca^2+^ imaging were fundamentally improved by obtaining well segmented CF varicosities (less than 10um) using Suite2p. The CF and the PC were regarded as a pair when they spatially overlap and their signals are significantly similar. The pair 1 and 2 in Figure 4 present excellent examples of CF-PC pair definition as the CF is surrounding the PC dendrite and their Ca^2+^ activity also show robust correlation. Employing the strategy, we re-analyzed the entire data set, showing that the signal correlation and the amplitude correlation are significantly high between the CF-PC pairs. In response to the argument pointing out that CF-PC correlation is not significantly different between paired and neighbor unpaired, they now show significant difference with the data analyzed with Suite2p (Figure 4D) and it is in line with the population synchrony result that CF correlation falls as a function of mediolateral separation (Figure 1F).

3) Quantification of the transients is also a problem. It is entirely unclear how events are detected after non-negative deconvolution and denoising (which may be subject to caution given the signal to noise of the recordings and the possibility of overfitting, see Figure 1—figure supplement 1B). Failure to detect close-by events may arise in the report of artificially large event amplitudes. When multiple events are detected does the amplitude correspond to the summed peak of the DF/F trace or to the actual individual events amplitudes?The very large amplitude of the CF transients (more than 2000 % in Figure 1) is surprising, as it may exceed the dynamic range of GCaMP6, particularly considering that ROIs cover large regions without any signal. Baseline subtraction, which may explain these large numbers if excess subtraction is applied, appears to be a recurrent issue in the paper with some panels displaying very low background noise (RFP in Figure 4—figure supplement 1A) and other very high (GFP in the same figure and in panel 1B).Furthermore, the signals appear saturated in many of the figures as well as in the movies, which does not really allow to judge the dynamic range and shape of the activated elements. The two movies have compression issues which do not allow clear visualization on a frame by frame mode.Finally, data of Figure 4 and the two movies originate from the same field of view and the two movies represent the same 20s but with very different levels of background subtraction and saturation. It is hard to understand why the same data are presented twice, and why not provide a more extensive sample of the data. Is it on purpose? If so, it should be explicitly mentioned.

In the new manuscript the Ca^2+^ event detection was performed with the state-of-the-art built-in deconvolution of Suite2p. The extracted Ca^2+^ signals were subtracted with background signals and then z-score normalized to avoid bias in amplitude quantification. The issue related to surprisingly large amplitudes of CF transients when CaImAn is employed was resolved with this approach (Suite2p and z-score normalization). Although the reviewer raised concerns regarding the saturation of figures and video, the data acquisition was carried out with high signal-to-noise level by minimizing laser power and properly adjusting gains. However, the images and movies were adjusted with contrast and brightness for clearer presentation of representative data. Please note that all analysis with Suite2p have been done with the original file. At this version of manuscript, we only present Video 1 for dual calcium imaging.

Reviewer #2:Roh et al. have revised their manuscript to sufficiently account for my previous concerns, most importantly by re-analyzing all of their calcium imaging data and enhancing the transparency of their data and methodological reporting.Notably, I still have some concerns over the role of movement in the amplitude of the reported signals. The authors correlate DF/F with paw speed, but not the amplitude of paw movement, or any other movements the animals make in response to airpuff stimulation. However, they now perform additional controls suggesting that imaging artifacts from bouton movement do not contribute to their results, and they have tempered their conclusions regarding the role of movement appropriately. Given the challenging nature of this problem, the level with which the authors have now addressed the role of movement seems reasonable. Likewise, the enhanced analysis of “independent” CF ROIs remains less than airtight, but in my opinion now meets a reasonable standard.Finally, there remain some typos, and some odd writing choices. Examples include statements like "this notion has been prevailed", and "sensory inputs disappear ex vivo". It is also atypical to end a manuscript with a final sentence telling the reader what the lab is working on next, rather than an overall statement about the significance of this study.

The typos were corrected and the manuscript was edited by native speakers. The final sentence was edited to finish by overall statement of the significance of this study.

Overall, however, the manuscript now convincingly makes an important link between the presynaptic activity of climbing fibers and the postsynaptic calcium transient in Purkinje cells in awake animals. It thus provides a significant advance, in parallel with recent work from the Christie lab, to our understanding of cerebellar learning mechanisms.